# Tbps wide-field parallel optical wireless communications based on a metasurface beam splitter

Yue Wu[1,4], Ji Chen [1,2,4] ✉, Yin Wang[2], Zhongyi Yuan[1], Chunyu Huang[3], Jiacheng Sun[3], Chengyi Feng[2], Muyang Li [1], Kai Qiu[3], Shining Zhu [3], Zaichen Zhang [1,2] ✉ & Tao Li [3] ✉

Optical wireless communication (OWC) stands out as one of the most promising technologies in the sixth-generation (6G) mobile networks. The establishment of high-quality optical links between transmitters and receivers plays a crucial role in OWC performances. Here, by a compact beam splitter composed of a metasurface and a fiber array, we proposed a wide-angle (~120°) OWC optical link scheme that can parallelly support up to 144 communication users. Utilizing high-speed optical module sources and wavelength division multiplexing technique, we demonstrated each user can achieve a communication speed of 200 Gbps which enables the entire system to support ultra-high communication capacity exceeding 28 Tbps. Furthermore, utilizing the metasurface polarization multiplexing, we implemented a full range wide-angle OWC without blind area nor crosstalk among users. Our OWC scheme simultaneously possesses the advantages of high-speed, wide communication area and multi-user parallel communications, paving the way for revolutionary high-performance OWC in the future.

Optical wireless communication (OWC) holds significant advantages, including a wide spectrum, high data rate, low latency, high security, low cost, and low energy consumption, to meet the stringent demands of 6G communications[1-3]. OWC facilitates the connection of end-user devices or sensors within the Internet of Things (IoT) network, enabling real-time communication, information sharing, and efficient handling of substantial data volumes[4,5]. The beam-steering technology, being capable of manipulating narrow collimated beam to different directions, is essential for realizing long-distance and high-data-rate OWC[6,7]. In traditional OWC systems, the beam steering function is usually conducted by SLMs or mechanically rotating mirrors[8-10]. Although the implementation methods are flexible, the beam steering angle is quite limited, due to the large diffraction pixel size or mechanical structure limitation. Moreover, the complex electrical or

mechanical components render these devices bulky and heavy, making them unsuitable for integration into highly portable devices. The recently developed concept of metasurfaces, a two-dimensional artificial material composed of sub-wavelength scale nano-structures, provides a promising solution for achieving beam steering devices with both a relatively large steering angle and a compact size.

Metasurfaces are capable of effectively manipulating multi-dimensional properties of light on an ultra-thin surface, including amplitude[11,12], phase[13-15], polarization[16-18], and orbital angular momentum[19,20]. Various compact optical functional devices have been realized based on metasurfaces, such as metasurface-based optical detectors[21-24], metalens-integrated cameras and microscopy[25-30], etc. In recent years, metasurfaces have also been employed to achieve compact beam steering devices with relatively large steering angles for

[1]National Mobile Communications Research Laboratory, School of Information Science and Engineering, Frontiers Science Center for Mobile Information Communication and Security, Quantum Information Research Center, Southeast University, 210096 Nanjing, China. [2]Purple Mountain Laboratories, 211111 Nanjing, China. [3]National Laboratory of Solid State Microstructures, College of Engineering and Applied Science, School of Physics, Nanjing University, 210023 Nanjing, China. [4]These authors contributed equally: Yue Wu, Ji Chen. ✉e-mail: jichen@seu.edu.cn; zczhang@seu.edu.cn; taoli@nju.edu.cn

optical wireless communication (OWC), thanks to their powerful light-field manipulation capabilities and sub-wavelength scale diffraction units[31–33]. However, the small structural size of optical metasurfaces makes it challenging to dynamically manipulate light pixel by pixel, as is easily done with radio frequency metasurfaces[34–36]. Consequently, a single optical metasurface can only achieve fixed-angle beam steering, leading to significant communication blind spots in a room. To address this issue, various dynamic control mechanisms have been introduced, including joint control methods involving metasurfaces and liquid crystals[37,38], double-layer metasurface counter-rotating methods[39], and relative movement between incident light and metasurfaces[40]. These works modulate a single incident beam into one or multiple dynamically deflected beams. However, since these output beams all originate from one single incident beam, they are not independent of each other and thus cannot be used to achieve parallel communication functions. In addition, the entire systems still require the assistance of complex mechanical or electronic devices, losing the high integration advantage of metasurfaces.

In this work, we proposed a compact beam splitter composed of a fiber array and metasurface, enabling up to 144 end-users to independently carry out information transmission. By employing 25G high-speed optical modules as the signal source, combined with wavelength division multiplexing technology, we demonstrated that each channel can achieve a communication rate of up to 200 Gbps. Consequently, the communication capacity of the entire system can exceed 28 Tbps. The dimensions of both the fiber array and the metasurface are compact, measuring 1.2 × 1.2 mm, making the core components of the device space-efficient. In addition, through the wide-angle (~120°) and polarization multiplexing design of metasurface, full coverage and no interference between different end-users can be achieved. Our metasurface-based beam splitter not only circumvents the difficulty of dynamic beam control in a compact device but also suggests an effective solution for future high-quality OWC that simultaneously possesses the advantages of wide area, multiple end-users, no blind spots, and ultra-high speed.

## Results

### The metasurface beam splitter

The core component of the beam splitter is depicted in Fig. 1a, which features a two-dimensional (2D) fiber array (Reful, RH-S-12-12-100-100-1m-FCAPC) integrated with a metasurface. By connecting optical signals to different fibers, the light emitted from the end face of the fiber illuminates different parts of the metasurface. Subsequently, the manipulated light would be deflected to various angles to achieve target communication. The assembling of fiber array and metasurface has been successfully implemented to realize collimated beam steering in light detection and ranging (LiDAR)[41]. However, to achieve full area coverage in OWC, it is necessary to systematically analyze the light field emitted from the fiber and its modulation effect by the metasurface. The divergence of modulated beam strongly depends on the distance between the fiber array and metasurface. For a given distance, optical beams will be modulated into well-collimated beams, rendering them suitable for high-speed and relatively distant OWC scenarios. First, we derive the phase profile of the metasurface and the appropriate distance based on the fiber output light field distribution.

The electric field of light emitted from a fiber port can be theoretically approximated by a Gaussian beam model, expressed as

$$E(r,z) = A_0 \frac{\omega_0}{\omega(z)} \cdot e^{-\frac{r^2}{\omega^2(z)}} \cdot e^{-i\left[k_0\left(z + \frac{r^2}{2R(z)}\right) - \tan^{-1}\left(\frac{z}{z_0}\right)\right]}, \quad (1)$$

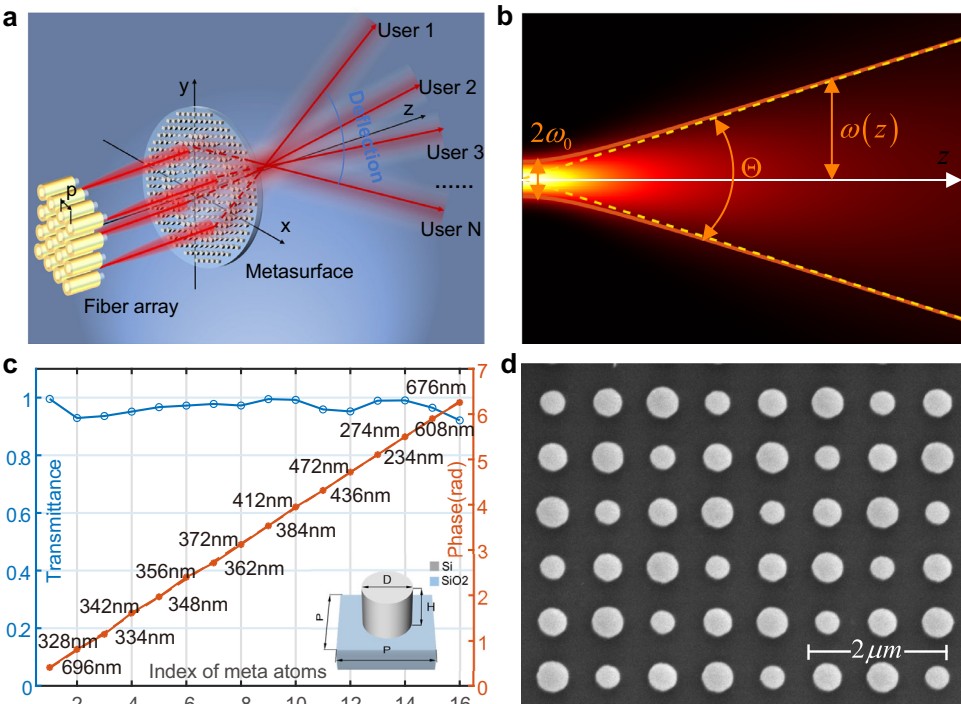

**Fig. 1 | Design of the metasurface beam splitter. a** Schematic of the core component of 2D metasurface beam splitter, *p* is the periodic distance of fibers in the fiber array. Note that, for clarity of presentation, the distances, dimensions, etc. are not actual scale. **b** Schematic of the Gaussian beam model for fiber output light field. *Θ*, $\omega_0$ and $\omega(z)$ are the divergent angle, beam waist and beam radius at z, respectively. **c** Phase (brown stars) and transmittance (blue circles) of meta-atoms with 16 different structural parameters, simulated by FDTD solutions at the wavelength of 1550 nm. The diameters of the nano-posts are marked along the phase distribution line. Inset is the schematic of metasurface unit cell with period P, silicon nano-cylinder height H and diameter D on a SiO₂ substrate. **d** Top-view scanning electron microscopy (SEM) image of part of the fabricated metasurface.

where, $A_0$ is the amplitude of the electric field, $z$ is the distance in propagation direction, $z_O$ is the Raileigh length, $r$ is the radial distance from the center axis, $\omega_0$ is the beam waist, $\omega(z)$ and $R(z) = z[1 + (\pi\omega_0^2/\lambda z)^2]$ is the beam radius and the radius of curvature at $z$, respectively, as shown in Fig. 1b. The second exponential term in the equation represents the phase profile of the Gaussian beam. Through experimental characterization of beam radius at different distances $z$, the divergence angle $\Theta$ of Gaussian beam can be obtained. Consequently, $\omega_0$, $\omega(z)$, and $R(z)$, can be derived successively (see Supplementary Note S1 for details). In this work, the center working wavelength is set as 1550 nm, which is commonly used in optical communications. According to the experimental characterization results, the derived divergence angle $\Theta = 0.094$ rad, the corresponding beam waist $\omega_0 = 5.27$ μm.

If a metasurface with phase $\varphi_{lens}$ is placed in front of the fiber with a distance of $d_O$, the phase profile just before the metasurface would be

$$\varphi_1(r) = k_0\left(d_0 + \frac{r^2}{2R(d_0)}\right) - \tan^{-1}\left(\frac{d_0}{z_0}\right) \quad (2)$$

where, $k_0 = 2\pi/\lambda_0$ is the wavenumber of 1550 nm wavelength beam. Since the terms $k_0 d_0$ and $-\tan^{-1}(d_0/z_0)$ are all constant independent of $r$, the phase profile can be simplified as a quadratic phase form shown as $\varphi_1(r) = k_0 r^2/2R(d_0) + C$. When the divergent beam passes through the metasurface at position $r_0$, it would deflect to certain angles. The deflect angles are determined by the generalized Snell's law[42], which would be expressed as,

$$\sin\theta(r_0) = \frac{1}{k_0}\frac{d}{dr}(\varphi_{lens} + \varphi_1) = \frac{1}{k_0}\frac{d}{dr}\left(\varphi_{lens} + \frac{k_0(r - r_0)^2}{2R(d_0)}\right) \quad (3)$$

where, $d/dr$ denotes the symbol for the derivative with respect to $r$. To manipulate the divergent beam into collimated beam, $\sin\theta$ should be a constant, which results in the manipulated phase $\varphi_{lens} + \varphi_1$ only contains the linear term with respect to $r$. Thus, we can obtain the metasurface phase is

$$\varphi_{lens}(r) = -\frac{k_0 r^2}{2R(d_0)} \quad (4)$$

In this way, the beam steering angle at position $r_O$ of the metasurface would be $\sin\theta = -r_0/R(d_0)$. In this work, the fiber array contains $12 \times 12$ fibers with intervals of 100 μm. To realize a wide coverage with beam steering angle of $\pm 60°$, the diameter of the metasurface and the distance $d_0$ is determined as 1.725 mm and 900 μm, respectively. (see Supplementary Note S2 for more details on parameters setting)

To efficiently modulate the optical field, we employ a propagation phase type metasurface, which was constructed by silicon cylindrical nano-posts with a height of 1000 nm and varying radius of round cross-sections, on a SiO₂ substrate. Inset in Fig. 1c shows the schematic of the metasurface unit cell. Through finite-difference time-domain (FDTD) simulation, 16 kinds of nano-posts with different cross-section radius were identified at the wavelength of 1550 nm, which cover 0–2π phase range and all exhibit over 90% transmittance, as shown in Fig. 1c. The metasurface was fabricated using standard electron-beam lithography (EBL) and dry etching (see "Methods" for details). Figure 1d shows the top view of a scanning electron microscope (SEM) image of the metasurface structure.

## Characterization of metasurface beam splitter performances

The performances of beam splitter were characterized at different distances and deflection angles relative to the metasurface, the experimental setup of which is shown in Fig. 2a. Light modules with different wavelengths were chosen as the light source and efficiently coupled into a fiber (experimentally measured efficiency is approximately 87%). The distance between the fiber array and the metasurface is precisely controlled to be equal to $d_0$, ensuring the generation of a collimated output beam. The metasurface is mounted on a two-dimensional stage, enabling control of relative movement between the fiber array and metasurface along the x and y directions. To analyze the deflected light spots at various z distances and deflection angles, a beam spot analyzer (NanoScan 2s, Ophir) is employed. Before characterizing the beam manipulation performance, all the 144 fibers are connected with the light sources and the emitted light beams are all modulated by the metasurface. Figure 2b shows the results of these modulated beams detected by the infrared laser viewing card.

The initial characterization focuses on the metasurface's performance in modulating a divergent light beam into a collimated beam. Figure 2c shows the comparison of measured intensity distributions of the light beam at 0° with respect to different distances after the fiber facet without (first row) and with (second row) the manipulation of metasurface. It is observed that, in the absence of metasurface manipulation, the beam exhibits rapid divergence. In comparison, the light beam modulated by the metasurface maintains its shape even at considerable distances (after a propagation of 1 m, the beam radius remains smaller than 5 mm). The curves depicting the beam radius in relation to distance z for the divergent and collimated beams are presented in Fig. 2d. Utilizing linear regression, the calculated divergent angles for these two beams are found to be 5.36° and 0.25°, respectively. This outcome underscores the powerful manipulation capability of the metasurface in transforming a divergent light beam into a precisely collimated one.

The beam steering performance of the metasurface is then characterized by testing the beam spots shape and deflection efficiencies at different angles. Figure 2e shows the intensity distributions of beam spots for deflection angles ranging from 0° to 60°. Notably, even at a deflection angle as large as 60°, the beam spot still maintains relatively good quality. Figure 2f shows the deflection efficiencies of five wavelengths (1560.61 nm, 1552.52 nm, 1550.12 nm, 1546.92 nm, and 1539.77 nm) with respect to different deflection angles, which is defined as the ratio of deflected beam power to input beam power. The 1550.12 nm is the center wavelength used in the wavelength division multiplexing (WDM) communication in the "Characterization of metasurface-based OWC system" section. The 1552.52 nm and 1546.92 nm wavelengths are the two boundary wavelengths for relatively long-distance (~2.6 m) WDM communication, while the 1560.61 nm and 1539.77 nm wavelengths are the two boundary wavelengths for relatively short-distance (<1 m) WDM communication (see Supplementary Note S3 for detailed analysis). The results show that for all five wavelengths, the deflection efficiency maintains greater than 40% until the angle exceeds 60°, ensuring high-speed OWC through WDM in a wide-angle range.

## Characterization of metasurface-based OWC system

In previously reported studies, reflective metasurfaces designed on the principle of holography have been utilized to deflect multiple beams from one incident light for broadcasting in optical wireless communication (OWC) systems[6,38]. However, in these works, all the deflected output beams carry identical information, unable to achieve parallel communication for multiple users. In our research, we adopt a different approach: loading different optical signals onto distinct fibers within an array. Then the signals would be steered by the metasurface to corresponding users, resembling a space division multiplexing (SDM) scheme. Consequently, the inherent capability to deliver distinct information to different users is achieved. This opens avenues in our work for employing other multiplexing schemes, such as WDM, to significantly enhance communication capacity for each individual user. The metasurface is designed for a working wavelength of 1550 nm. Due to the chromatic aberration property of the metasurface, when the wavelength of the incident light varies, the deflection angle

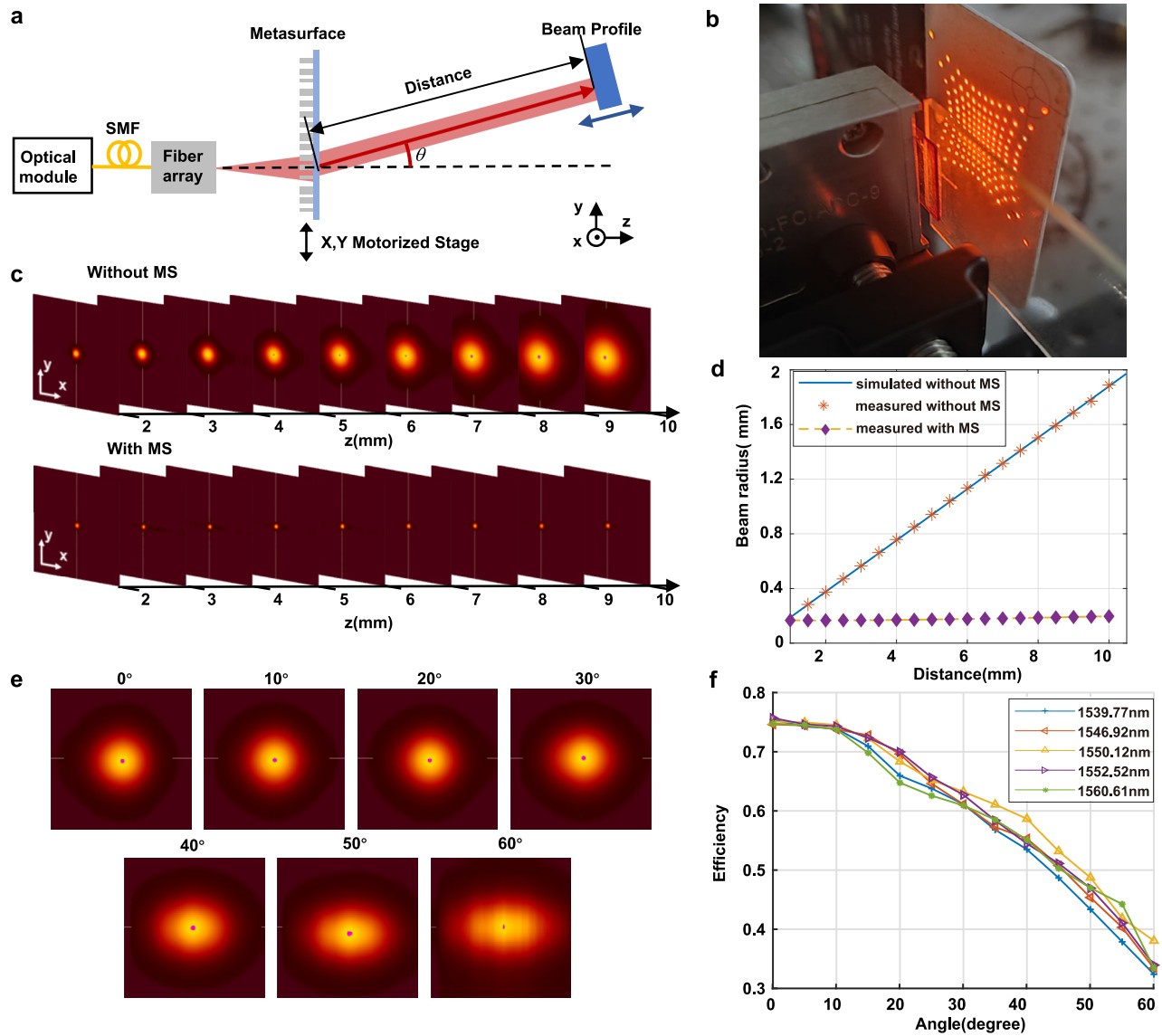

**Fig. 2 | Characterization of metasurface beam splitter performances.**
**a** Schematic of the experimental setup. **b** The 144 manipulated beams detected by the infrared laser viewing card. **c** The captured intensity distributions at different distances after the fiber facet with beam steering angle of 0°, without (first row) and with (second row) the manipulation of metasurface (MS). **d** The corresponding measured beam radius. **e** The captured beam profiles of different beam steering angles when distances from the metasurface were fixed as 15 cm. **f** Measured deflection efficiency of five wavelengths (1560.61 nm, 1552.52 nm, 1550.12 nm, 1546.92 nm, and 1539.77 nm) at different beam steering angles.

of the output light modulated by the metasurface will also alter. Therefore, to use WDM for each communication channel, a comprehensive analysis of multiple factors is required, including the multiplexed wavelengths, deflection angles, and communication distance, in order to determine the optimal parameters of the communication system (for detailed analysis, see Supplementary Note S3).

Figure 3a shows the experimental setup and layout for metasurface-based OWC system. Eight 25-Gbits dense wavelength division multiplexing optical modules (DWDM-SFP25G-10, Cisco) operating within wavelengths ranging from 1546.92 nm to 1552.52 nm (specifically: 1546.92 nm, 1547.72 nm, 1548.51 nm, 1549.32 nm, 1550.12 nm, 1550.92 nm, 1551.72 nm, and 1552.52 nm) were driven by a pseudo-random binary sequence generator from the evaluation board (FPGA KCU116, Xilinx). The chosen wavelengths are the eight closest ones to the center wavelength of 1550 nm in WDM communication, which ensures that each channel achieves a high-speed communication rate while maintaining a relatively long communication distance. Each optical module outputs a beam carrying a 25 Gbps on-off-keying

(OOK) pseudo-random binary sequence (PRBS) with a length of $2^{31}$-1. These beams were then combined using a wavelength division multiplexer (FMU-D402160M3, FS) and amplified by an erbium-doped fiber amplifier (EDFA-1, AEDFA-BO-23-B-FA, Amonics). The resultant composite beam, carrying total 200 Gbps signal, would be introduced into one fiber of the fiber array, emitting from the fiber array end facet, and modulated by the metasurface to certain angles. After propagating in free space for 50 cm, the optical signals were received and coupled into a single-mode fiber by a collimator and demultiplexed by the wavelength division demultiplexer. To compensate for the optical power loss of the receiving system, EDFA-2 (AEDFA-33-B-FA, Amonics) was employed to amplify the signals in each of the WDM channels for photodiode detection of the optical modules and subsequently characterized by the bit error rate tester (BERT).

Figure 3b shows the measured bit-error-ratio (BER) curves versus the optical received power (ORP) for the eight wavelengths at a deflection angle of 30° and a receive distance of 50 cm. It is observed that all eight wavelengths exhibit similar performance. It is because the

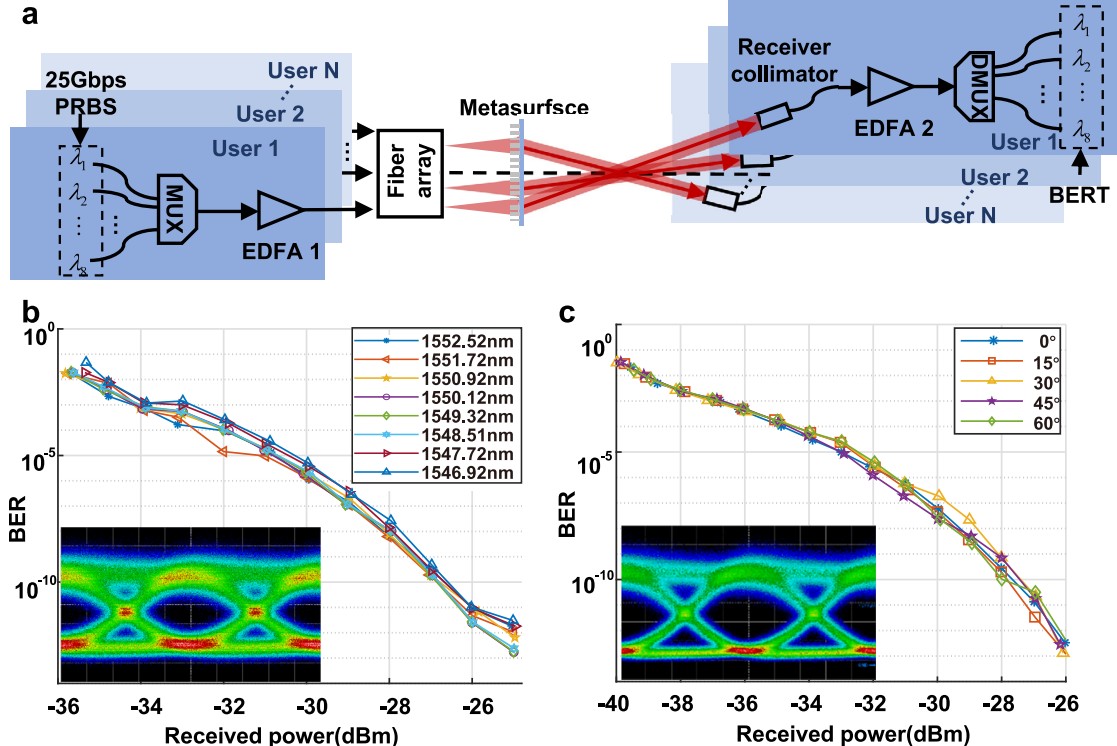

**Fig. 3 | Experimental setup and measured results of the metasurface-based OWC system. a** Experimental setup schematic of the metasurface-based OWC system. EDFA erbium-doped fiber amplifier, PRBS pseudo-random binary sequence, BERT bit error rate tester, MUX multiplexer, DMUX demultiplexer. **b** The measured BER-ORP (bit-error-ratio versus optical received power) curves for eight spaced wavelengths from 1546.92 to 1552.52 nm, each of which carries 25 Gbits signals. Inset: Receiver eye diagram at the wavelength of 1550.12 nm at −30 dBm. **c** BER curves versus ORP at different deflection angle at the wavelength of 1550.12 nm. Inset: the receiver eye diagrams for the deflection angle of 60° at −30 dBm.

maximum wavelength difference is about 5 nm which is three orders of magnitude smaller than the working wavelength for communication. Consequently, the influence of wavelength change can be disregarded, ensuring the high-speed communication capability of each channel in our OWC system. The inset in Fig. 3b is the receiver eye diagram at the center wavelength of 1550.12 nm and a power of −30 dBm. Figure 3c shows the measured BER-ORP curves for deflection angles ranging from 0° to 60° at a wavelength of 1550.12 nm. The inset shows the eye diagram obtained at the deflection angle of 60° and a received power level of −30 dBm. The open and clear eye diagrams indicate that the system could maintain good communication performances over a large angle range. A detailed discussion on PRBS signals at different wavelength channels is provided in Supplementary Note S4.

**High-speed, wide-field, parallel OWC system**

After characterizing the optical and communication performances, we constructed a prototype of a metasurface-based Optical Wireless Communication (OWC) system to demonstrate high-speed wide-field parallel communication. This developed OWC system has the capability of transmitting distinct high-definition (HD) video signals through two independent channels. Figure 4a illustrates the schematic diagram of the system, while the actual setup is depicted in Fig. 4b. The beam steering angles for the two information transmission channels were approximately 25° and 50°, respectively. The transmission process of video signals is identical in these two channels. Specifically, video signals were initially transmitted from the computers to the Module compliance board (MCB, DEB2-12M-A, DooDooTech), through data cables. Subsequently, the MCB controlled the optical module to convert video information into optical signals, which were then coupled into a single fiber of the fiber array. The optical signals emitted from the fiber facet were modulated by

the metasurface (shown as zoom-in picture 1 in Fig. 4b) and deflected to a specific angle. After propagating through free space for a distance exceeding one meter, the optical signals were captured by a receiver collimator (PAF2A-18C, Thorlabs) and transmitted to another MCB to transform the optical signal back into an electrical video signal. Through data cables, the HD video was finally displayed on the output monitor.

In channel 2, we not only demonstrated the functionality of video signal transmission but also the performance of dense wavelength division demultiplexing (DWDM) technology to improve information transmission capacity. Six wavelengths ranging from 1548.51 nm to 1552.52 nm were chosen as the optical carriers (C31-C36 DWDM-SFP25G-10, Cisco), in which $\lambda_1 = 1552.52$ nm carried the video signal displayed on PC 2 and $\lambda_2 \sim \lambda_6$ carried the five 25 Gbps PRGS 31 sequences simultaneously. Then the signals were combined by a wavelength division multiplexer (MUX, FMU-D402160M3, FS) and amplified by a EDFA (AEDFA-33-B-FA, Amonics) to pre-compensate for the power loss incurred by the MUX, DMUX and the collimator. After being captured by the receiver collimator, the optical signals were demultiplexed by the wavelength division demultiplexer (DMUX, FMU-D402160M3, FS). The zoom-in picture 2 in Fig. 4b shows the spectrometer detecting results of these six wavelengths optical signals after being multiplexed (light blue line) and before being demultiplexed (pale purple line). Signals carried by $\lambda_1$ were transformed back into electrical signal and displayed on monitor 2, while signals carried by $\lambda_2 \sim \lambda_5$ were received by optical modules driven by the evaluation board (FPGA KCU116, Xilinx) and characterized by the BERT, shown as zoom-in picture 3 in Fig. 4b. The 25 Gbps communication rates for each of the four wavelengths can be clearly seen. Signals carried by $\lambda_6$ were connected to the sampling oscilloscope (MP2110A, Anritsu) for eye diagram acquisition to

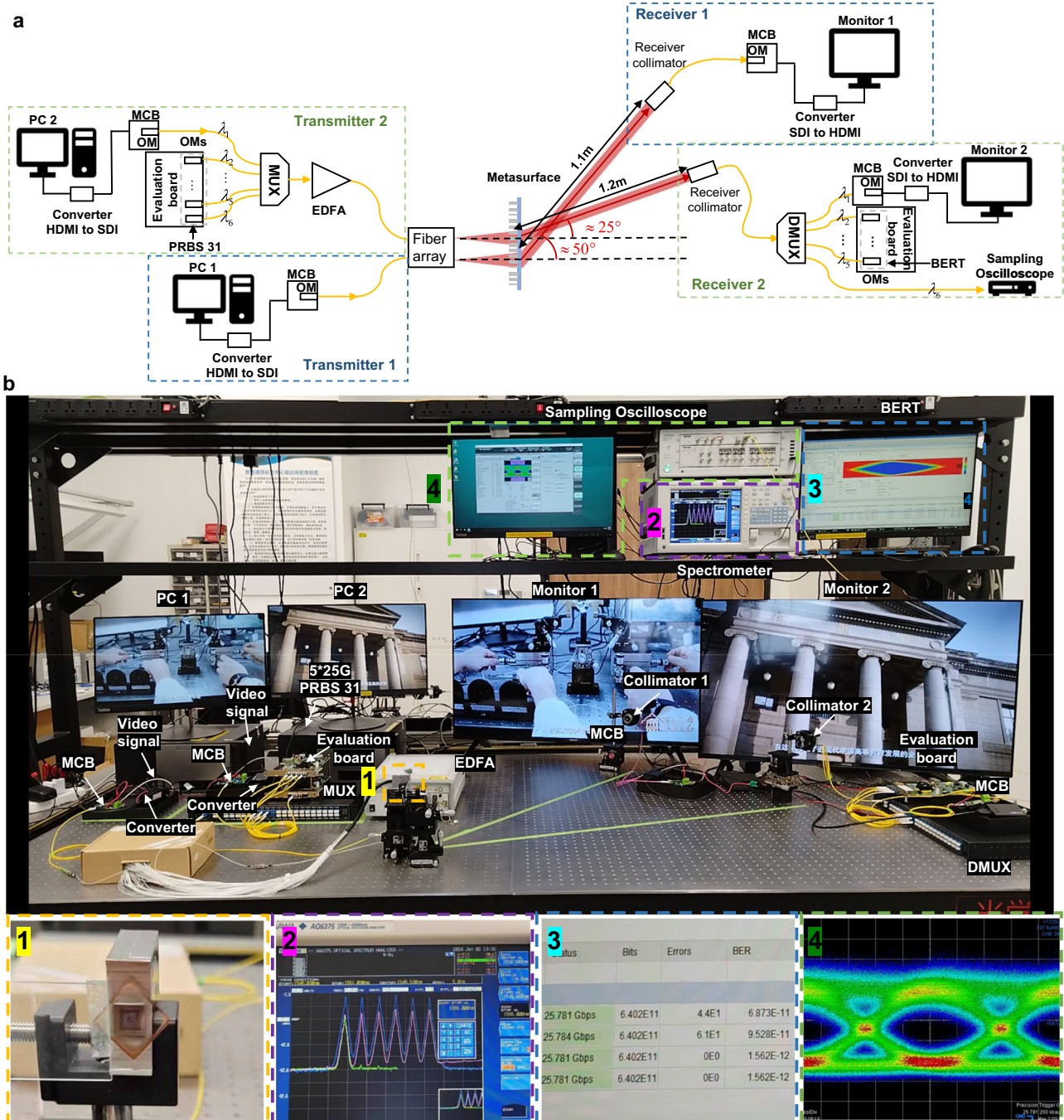

**Fig. 4 | Demonstration of the wide-field parallel OWC system. a** Schematic diagram and **b** the actual scene of the high-speed wide-field parallel OWC system. The main components in the system have all been labeled. The red and yellow dashed lines approximately represent channel 1 and channel 2, respectively. Zoom-in picture 1 shows the core component composed of the fiber array and a metasurface. Zoom-in picture 2 shows the detected wavelengths by the spectrometer, the light blue line represents the results detected after wavelength multiplexing, and the pale purple line represents the results detected before wavelength demultiplexing. Zoom-in picture 3 shows the bit error rate tester (BERT) results of signals carried by $\lambda_2 \sim \lambda_5$. Zoom-in picture 4 shows the eye diagram result of signals carried by $\lambda_6$. EDFA erbium-doped fiber amplifier, BERT bit error rate tester, MUX multiplexer, DMUX demultiplexer, MCB Module compliance board, OM optical module, PC personal computer, HDMI high-definition multimedia interface, SDI serial digital interface.

qualitatively evaluate the communication performance of the system, shown as zoom-in picture 4 in Fig. 4b. The comprehensive exhibition of the high-speed wide-field parallel OWC system and the detailed signal conversion processes during the information transmission, are shown in Supplementary Movies S1–S3. More details about the experimental setup and the power loss analysis of the system are provided in Supplementary Note S5.

## Design of full range coverage metasurface-based OWC

To enhance the performance of our OWC system, we further analyzed and designed a scheme capable of achieving full coverage of the communication area, enabling user devices to be positioned anywhere within the area. Since the radius of the collimated beam is hundreds of microns, even 144 beams are insufficient to cover the full range of the entire communication area. However, excessively

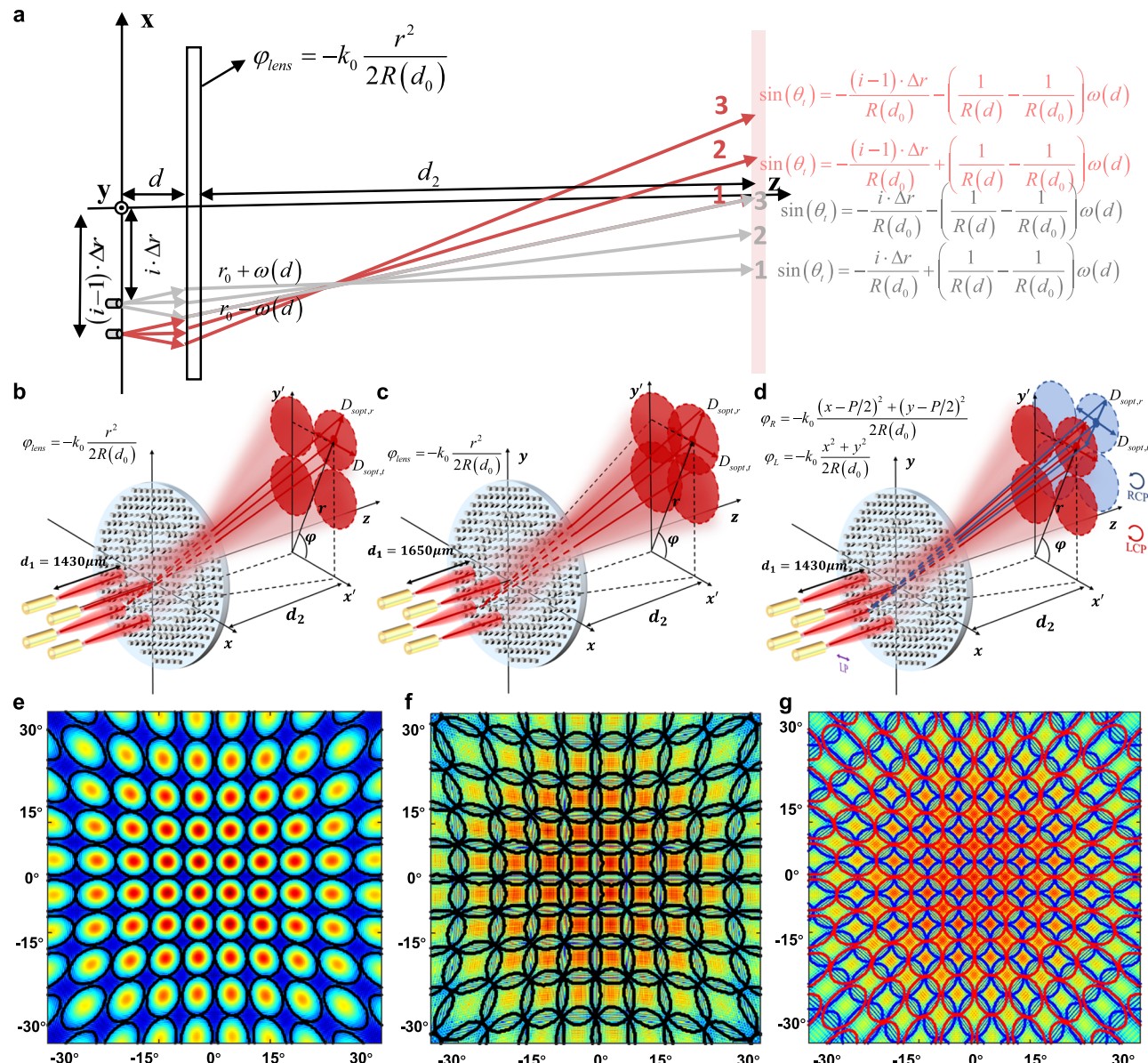

**Fig. 5 | The design of full range coverage metasurface-based OWC. a** Geometric relations of light rays from adjacent fibers, at critical full range coverage condition. The red lines represent the rays from fiber i-1, while the gray lines represent the rays from fiber i. The two angles are the beam deflection angle of line 1 from fiber i-1, and the beam deflection angle of line 3 from fiber i, respectively. The critical full-range coverage condition is to make these two deflection angles equal. The schematics of the same polarized beam coverage on the observation plane at $d_2 = 10$ mm, when $\Delta r$ in Eq. (5) is set equal to **b** $p$ and **c** $\sqrt{2}p$, the corresponding distances between the metasurface and the fiber array are 1430 μm and 1650 μm, respectively. **d** The schematic of LCP and RCP beam coverage on the same observation plane, when $\Delta r$ is set equal to $p$. **e–g** The beam intensity distributions on observation plane corresponding to the conditions in (**b–d**), respectively. The black wireframes in (**f**) show the boundaries of beams emitted from different fibers. The parts encircled by red wireframes and blue wireframes in (**g**) are the beam coverage areas from LCP incident light and RCP incident light, respectively.

large beam divergence would lead to a weakened light field and a significant reduction in OWC speed. Thus, the beam should be slightly divergent to meet the critical full coverage condition by precisely adjusting the distance between metasurface and the fiber array. There is a distance that satisfies the critical full coverage condition, in the case of $d > d_0$ and $d < d_0$, respectively. In Supplementary Note S6, these two situations are analyzed in detail. The case of $d > d_0$ was ultimately chosen due to its ease of implementation, the beam trajectories of which are shown in Fig. 5a. The gray and red lines represent the beam rays emitted from two adjacent fibers. To facilitate analysis, two rays at the edge and one ray in the middle of each fiber are studied, which are numbered as line 1, line 2, and line 3, respectively. As discussed in Supplementary Note S6, the critical

condition for full coverage is the steering angle of line 3 of $i$-th fiber ($\sin \theta_{i,3}$) equal to the angle of line 1 of $i - 1$ th fiber ($\sin \theta_{i-1,1}$), which were expressed as

$$-\frac{i \cdot \Delta r}{R(d_0)} - \left(\frac{1}{R(d)} - \frac{1}{R(d_0)}\right)\omega(d) = -\frac{(i-1) \cdot \Delta r}{R(d_0)} + \left(\frac{1}{R(d)} - \frac{1}{R(d_0)}\right)\omega(d)$$
$$\Rightarrow \frac{(R(d) - R(d_0)) \cdot \omega(d)}{R(d)} = \frac{\Delta r}{2}$$

(5)

where, $\Delta r$ is the spacing between two fibers in the fiber array. In this work, the utilized fiber array contains $12 \times 12$ fibers, with intervals $p = 100$ μm.

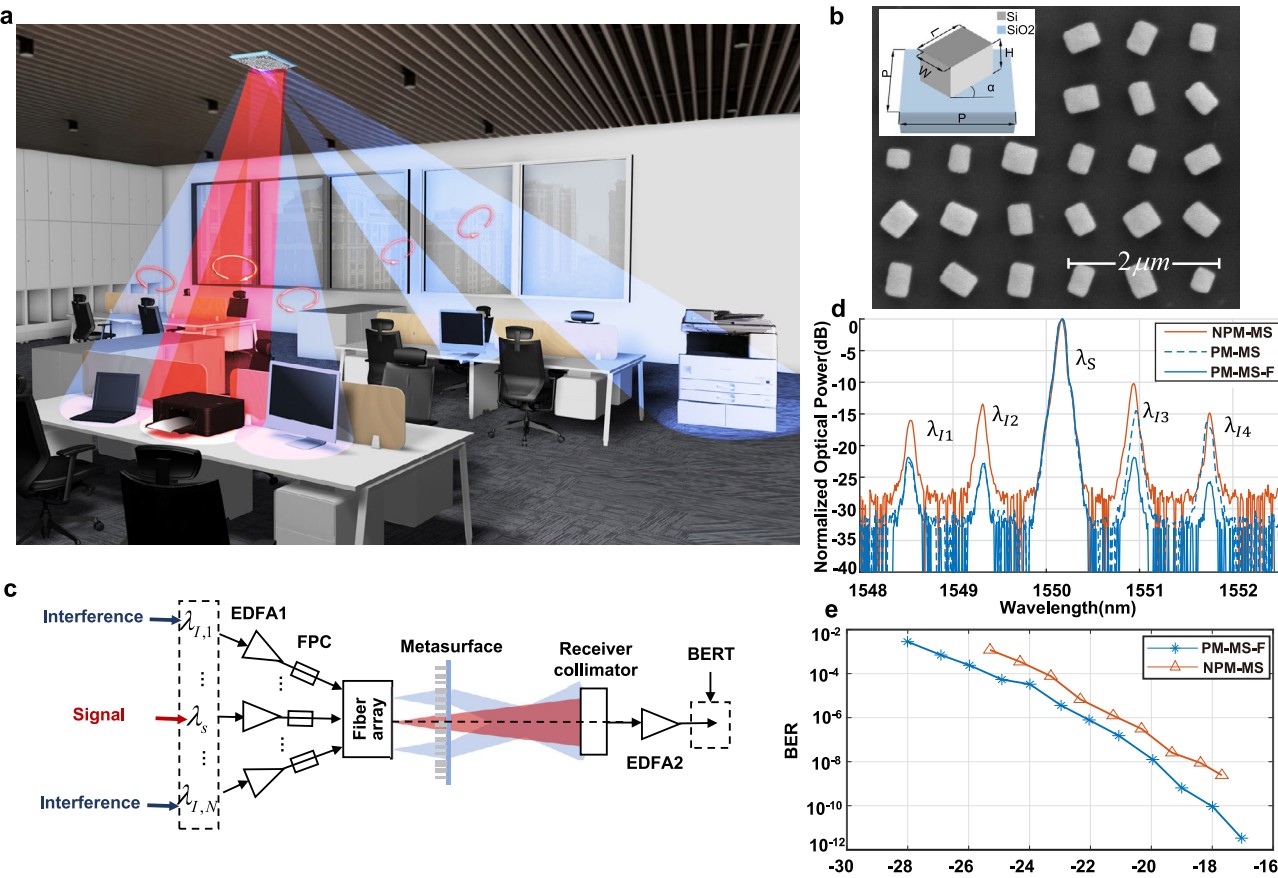

**Fig. 6 | Characterization of full range coverage OWC design. a** Working scenario schematic of the metasurface-based OWC. The three output beams on the left exhibit the full range coverage condition, while the two beams on the right exhibit the wide-filed coverage capability. It should be noted that the blue and red colors do not represent the wavelengths of the beam. Instead, they denote different rotations of circular polarization. **b** The SEM images of the polarization-multiplexed metasurface. Inset: schematic of the polarization-multiplexed metasurface unit cell with period of P, containing an amorphous silicon nano-brick with length L, width W, height H and orientations α patterned on a SiO₂ substrate. **c** Experimental setup for characterization of the full range coverage OWC performances. EDFA erbium-doped fiber amplifier, BERT bit error rate tester, FPC fiber polarization controller. **d** Received spectra of non-polarization-multiplexed metasurface method (solid brown line, abbreviated as NPM-MS), polarization-multiplexed metasurface method without polarization filtering (dashed blue line, abbreviated as PM-MS) and with polarization filtering (solid blue line, abbreviated as PM-MS-F), which were all normalized by the working signal power. $\lambda_s$ is 1550.12 nm, $\lambda_{I1}$-$\lambda_{I4}$ are 1551.72 nm, 1550.92 nm, 1549.32 nm and 1548.51 nm, respectively. **e** Measured BER-ORP curves for signals in polarization-multiplexed metasurface method with polarization filtering (PM-MS-F) and non-polarization-multiplexed metasurface (NPM-MS) method.

Through derivation in Supplementary Note S6, it is concluded that the spot shape of the light emitted from a fiber on the observation plane is an ellipse. Thus, if $\Delta r$ in Eq. (5) is set as the interval of nearest neighbor fibers, that is $\Delta r = p$, the spots distribution of the fiber array would be shown like Fig. 5b. In this case, although the spots of adjacent fibers are connected, there would still be considerable areas on the observation plane, that are not covered by light spots, as shown in Fig. 5e. Only when $\Delta r$ is set as the interval of second nearest neighbor fibers (as shown in Fig. 5c), that is $\Delta r = \sqrt{2}p$, the observation plane would be fully covered, and the spots of nearest neighbor fibers would overlap, as shown in Fig. 5f. As when communication, different fibers would transmit different signals, the overlap areas will bring crosstalk to end-users linking to adjacent fibers (see Section "Performances of full range coverage metasurface-based OWC" for details).

To address this issue, we introduced the polarization-multiplexed metasurface, which would exhibit different phase responses when illuminated with opposite circularly polarized light[26] or orthogonal linear polarized light[43]. The phase for modulating fiber output light into vacant area in Fig. 5b is obtained by applying a translation of $p/2$ in both $x$ and $y$ directions to the quadratic phase of Eq. (4), which would

be shown as,

$$\varphi_{lens}(r) = \frac{k_0[(x - p/2)^2 + (y - p/2)^2]}{2R(d_0)} \quad (6)$$

By manipulating the polarization of fiber array output light (as shown in Fig. 5d), the metasurface can achieve complete indoor beam coverage, as shown in Fig. 5g. In this case, $\Delta r$ is set equal to $p$, causing the illumination regions of identical polarized beams not to overlap. When communication, even the end-users are in overlapped regions, through polarization filtering, significant reduction of signal crosstalk would be achieved.

**Performances of full range coverage metasurface-based OWC**
Figure 6a illustrates the working scenario of the metasurface-based OWC. The metasurface beam manipulation device is installed on the ceiling, allowing the emission of beams at various steering angles to reach distinct locations within the room and received by end-user receivers. The metasurface is designed with wide-angle and polarization multiplexing properties, ensuring that the beams cover every corner of the room and eliminating communication blind spots. Each

emitted beam establishes an independent channel, enabling users to engage in parallel communications. This not only significantly enhances the overall communication capacity but also ensures the communication security of each channel. Besides, through polarization filtering during information reception, interference between adjacent regions can be effectively eliminated.

The polarizations of beams emitted from fibers can be manipulated by fiber polarization controllers (Thorlabs, FPC 900), which mechanically squeeze the optical fibers to induce birefringence. This process is analogous to employing a waveplate, with the ability to adjust both the polarization angle and delay continuously and independently. As a result, it facilitates the transformation of any input polarization state into the desired output polarization state.

The polarization multiplexing metasurface is employed to realize full range coverage. The inset in Fig. 6b shows the schematic of polarization multiplexing metasurface unit cell, which has the same period $P$ and height $H$ but different length $L$, width $W$, and orientation $\alpha$. The sizes and orientations determine the phase responses of the unit cells, with respect to different circular polarizations. In this work, the unit cells were fabricated of silicon material on silica substrate, with period $P$ and height $H$ set as 800 nm and 1000 nm, respectively. The FDTD Solution was used to simulate the amplitude and phase responses of metasurface unit cell with respect to different $L$ and $W$ to determine the satisfied structure parameters. More details on determining the structure parameters are shown in Supplementary Note S7. The fabrication process is shown in the "Methods" section. Figure 6b shows the SEM image of the polarization-multiplexed metasurface.

As discussed in the preceding section, the full range coverage condition can be realized by both polarization-multiplexed metasurface and non-polarization-multiplexed metasurface. In these two methods, the adjacent light spots originate from the emitted light of the nearest and second nearest neighboring fibers in the fiber array. By substituting $\Delta r = p$ and $\Delta r = \sqrt{2}p$ into Eq. (5), the corresponding distances between the metasurface and the fiber array $d_1$ in these two methods can be obtained, which are 1430 μm and 1650 μm, respectively. To demonstrate the polarization-multiplexed metasurface possesses better communication performance at full range coverage condition, we established a characterization setup, as shown in Fig. 6c. A working signal with wavelength $\lambda_s$ of 1550.12 nm was transmitted through a fiber. For the convenience of measuring the impact of noise from adjacent beams, four interfering signals with carrier wavelengths $\lambda_{I1}$-$\lambda_{I4}$ of 1551.72 nm, 1550.92 nm, 1549.32 nm and 1548.51 nm, respectively, were also loaded into the adjacent fibers. The receiver is positioned behind the metasurface at the working signal beam region to capture all the signals and transmit them to a spectrometer for analysis. For non-polarization-multiplexed metasurface, the communication performances were characterized directly. For polarization-multiplexed metasurface, the incident beams from fibers were all set as linear polarized (LP) light. Then the incident beams would be modulated to both left-handed circular polarization (LCP) beams and right-handed circular polarization (RCP) beams by the metasurface. The working signal beam was set as an LCP beam, thus the LCP and RCP beams surrounding the working signal beam region would all contribute to the interference signals. When the signals were received, an LCP polarizer could be employed to effectively filter part of the interference signals. Figure 6d shows the received spectra of non-polarization-multiplexed metasurface (solid brown line) method, polarization-multiplexed metasurface method without polarization filtering (dashed blue line) and with polarization filtering (solid blue line), respectively. It can be observed that the polarization-multiplexed metasurface method with filtering exhibits much lower interference signals intensity and higher signal-to-noise plus interference ratio (SNIR), which is more than 10 dB improvement compared with that in non-polarization-multiplexed metasurface method. Figure 6e shows the BER-ORP curves in these two methods. It can be seen that, at an equivalent received power level,

the BER of results in polarization-multiplexed metasurface method is about one order of magnitude lower than that in non-polarization-multiplexed metasurface method. The notable distinction between these two curves indicates that our polarization-multiplexed metasurface exhibits superior communication performances, at full range coverage condition. A detailed comparison of the SNIR performances of non-polarization-multiplexed metasurface method and polarization-multiplexed metasurface method with and without polarization filtering is shown in Supplementary Note S8.

## Discussion

For the issue of angular modulation of metasurface, as the deflection angle increases, the efficiency of the deflected beam decreases obviously. This efficiency decrease is attributed to the challenges in supplying necessary transverse momentum for large-angle beam deflection by traditional metasurface structures, such as periodic cylinders or cuboids. Consequently, only a fraction of the energy is converted into the target deflected beam, with the remaining either passing through the metasurface directly or being converted into losses. Significantly, the metasurface in this work effectively accomplishes large-angle beam deflection with relatively good beam quality, which also maintains fair efficiency. If the efficiency of deflecting wide-angle beams can be further improved, it will effectively enhance the performance of metasurface-based OWC. There are two main methods to improve the efficiency of large-angle beam deflection. One involves structural optimization using techniques such as topological optimization[44,45] or deep learning[46–48], targeting large-angle deflection efficiency as the optimization objective and using structure shapes or positions as parameters. This can result in free-form structures or non-periodic arrangements with high modulation efficiency for large-angle beams[49]. The other type of method involves designing tilted metasurface structures tailored to each deflection angle to enhance efficiency[50]. Both types of methods require comprehensive consideration of fabrication precision and structural stability, necessitating further systematic research. For the issue of communication rate limit in our work. The communication rate per channel has an upper limit because of the chromatic property of metasurface, which is explained in detail in Supplementary Note S3. Theoretically, the number of communication channels can be increased without constraint, just by increasing the number of fibers. However, in practice, more communication channels require larger size of fiber array and metasurfaces, which would be limited by the processing technology. Furthermore, in experiment, for the ease of precisely adjusting the distance between the metasurface and the optical fiber, we employ a translation stage for movement. In practical devices, once the distance between the metasurface and the optical fiber is determined, the components can be secured and assembled using optical clear adhesive tape or 3D-printed molds.

OWC offers advantages such as electromagnetic interference resistance, strong confidentiality, high energy efficiency, abundant spectrum resources, and large communication capacity. These characteristics are advantageous in densely populated environments such as in open office rooms, conference halls, exhibition halls, airplane cabins, train cabins, etc[51]. By integrating with indoor high-precision optical sensing, the OWC technology will give rise to a series of intelligent indoor applications, such as health monitoring, fall detection, person identification, etc[52]. To achieve these functionalities, continuous developments are also needed in various technologies, such as light sources, beam modulation technology, target positioning technology, and optical signal receivers. Additionally, it requires comprehensive optimization methods for the layout of the entire communication system, energy distribution, and system losses. For example, intelligent real-time management of wireless channel according to communication demands[53], or solutions to the line-of-sight blockage problem between a base station and a user[54].

Metasurfaces, with their ultra-light and ultra-thin advantages and flexible design capabilities, will have broad application prospects in the realization of highly integrated and high-performance optical communication devices, such as wide-angle optical signal receivers, high-precision positioning devices, and intelligent reflecting surfaces (RIS) for channel optimization[55].

In conclusion, based on the integration of metasurface and fiber array, we have realized a high-speed, wide beam steering angle, and multi-channel parallel transmission OWC system. The system directs light signals into different fibers within the fiber array, then the light is emitted and illuminated on different regions of the metasurface, and modulating to users at different angles. Throughout this process, each channel operates in parallel and independently, allowing the entire device to function as a highly compact spatial multiplexer with 144 channels. The silicon metasurface, based on the principle of propagation phase modulation, provides the capability for large-angle beam steering and high modulation efficiency. Experimentally verified metasurface bandwidth in the tens of nanometers ensures that each channel can support the operation of 25 G optical modules with up to eight wavelengths. This wavelength division multiplexing function enables each channel to achieve a communication rate of up to 200 Gps, resulting in an overall device capacity exceeding 28 Tbps, which is the highest reported communication rate in beam steering-based OWC works to date. The detailed performance comparisons of our work and other relevant works are shown in Supplementary Table S1. Furthermore, we explored the fine adjustment of the distance between the fiber array and the metasurface to transform perfectly collimated beams into slightly divergent beams, achieving full coverage of indoor optical beams. Leveraging the unique polarization reuse characteristics of the metasurface, we achieve indoor full coverage without interference between adjacent users. Our OWC system maximizes the advantages of the metasurface's ultra-light and ultra-thin dimensions, as well as its efficient control over light phase and polarization. This effectively addresses limitations in existing OWC systems related to beam deflection angles, user numbers, and communication rates, providing a brand-new approach for future high-performance wireless optical communication.

## Methods

### Numerical simulation
The transmission coefficients and electromagnetic field distributions of the metasurface unit cell were simulated with the finite-difference time-domain method via commercial software from Lumerical Inc., FDTD Solutions. Periodic boundary conditions were set in the x and y directions to create an infinite periodic array of the unit cell, as depicted in Figs. 1c and 6b. Perfect matching layers are employed along the z-axis to absorb the outgoing waves. The refractive indices of fused silica substrate and amorphous silicon were from the Handbook of Optical Constants of Solids (E.D. Palik).

### Fabrication of the metasurfaces
To fabricate the sample, first, a 1-µm-thick amorphous silicon film is deposited on the SiO₂ substrate using Plasma enhanced chemical vapor deposition (PECVD). 225 nm PMMA A4 resist film and a 50-nm-thick layer of a water-soluble conductive polymer (AR-PC 5090) were spin-coated onto the substrate in sequence and loaded into the electron-beam lithography system (Elionix ELS-F125). After the exposure, the conductive polymer was then dissolved in water, and resist was developed in a resist developer solution (MIBK: IPA = 1: 3) for 120 s and fixer (IPA) for 60 s in sequence. A 40 nm electron thermal evaporated chromium layer was used to reverse the generated pattern with a liftoff process and was then used as a hard mask for dry etching the silicon layer. Finally, the sample is dry-etched and immersed in the stripping solution (ceric ammonium nitrate), and the chromium layer is removed from the substrate, leaving only the desired metasurface structure on the substrate.

## Data availability
The data that support the findings of this study are provided in the Supplementary Information/Source Data file. Source data are provided with this paper.

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

## Acknowledgements

This work is financially supported by the National Key R&D Program of China (2022YFA1404301), National Natural Science Foundation of China (Nos. 12104223, 62325504, 61960206005 and 61871111), Jiangsu Key R&D Program Project (No. BE2023011-2), Young Elite Scientists Sponsorship Program by CAST (No. 2022QNRC001), the Fundamental Research Funds for the Central Universities (Nos. 2242022R10128 and 2242022k60001), and Project of National Mobile Communications Research Laboratory (No. 2024A03).

## Author contributions

J.C., Y.Wu, Z.Z. and T.L. developed the idea. Y.Wu proposed the design and performed the numerical simulation. Y.Wu and J.C. conducted the optical testing experiments with the assistance of Z.Y., Y.Wu and Y.Wang conducted the optical communication experiments with the assistance of C.F. Y.Wu and C.H. fabricated the samples with the help from J.S., M.L. and K.Q. J.C. and Z.Z. supervised the project. J.C. and Y.Wu analyzed the results and wrote the manuscript under the guidance of S.Z. All authors contributed to the discussion.

## Competing interests

The authors declare no competing interests.
