## [Peer Review File · Nature Communications]

Tbps wide-field parallel optical wireless communications based on a metasurface beam splitterREVIEWER COMMENTS

Reviewer #1 (Remarks to the Author):

The manuscript presents an optical wireless communication (OWC) link scheme that aims to simultaneously provide high-speed, multi-user information transfer over a wide area with no blind spots.

The authors realize this scheme by combining a fiber array and a beam steering metasurface, experimentally demonstrating 200 Gbps data transmission, 60° deflection, and complete range covering through

polarization multiplexing, and a prototype OWC for high-definition video transmission.

The results presented are an improvement to the state-of-the-art beam steering device's deflection angle and data rate. At the same time, the approach permits independent and parallel information transmission with no blind spots. The manuscript is well structured, and the data presented

is of good quality. There is sufficient information to reproduce the results, and the work is of interest to a broad audience.

Major remarks:

1. The introduction states that a limitation of single metasurfaces for OWC is fixed-angle beam steering, which is presently mitigated by control mechanisms such as liquid crystals or optomechanics that allow dynamic beam steering. Dynamic beam steering is engaging in OWC as it provides information that can be transferred to moving users. Here, Fig. 6a shows information transfer to a walking person and a robotic vacuum cleaner with the proposed scheme. However, this work

describes a single metasurface setup for which communication lines with moving users should not be possible. The authors should describe the mechanism by which connections to moving users can be established or revise Fig. 6a to avoid misleading readers on the capabilities of the proposed link scheme.

2. In line 169, the authors state that at 60° deflection angles, the beam spots retain high quality. However, Fig. 2e shows considerable deformation of the beam profile, and Fig. 2f shows the

60° beam to carry less than half the energy of the 0° beam. Therefore, the statement appears to contradict the data. These statements repeat in the discussion, as lines 368-369 subjectively

state that "the metasurface in this work effectively accomplishes large-angle beam deflection with good beam quality, which also maintains relatively high efficiency"

Minor remarks:

1. In Equation 1, the quantity A_0 should be described in the text as the other values are.

2. There is inconsistent notation between equations 1-2, the text of the manuscript, and the supplementary material when referring to \tan^{-1} and \arctan . A consistent notation should be used

3. In Equation 2, the quantity k_0 should be described in the text.

4. In Equation 3, the quantity d should be described in the text.

5. The left axis of Fig. 1c is labeled "Efficiency"; however, the caption and line 138 refer to the

panel as plotting "transmittance". Thus, the panel's axis label should be changed.

6. In line 145 the authors should quantify the efficiency of light coupling into the fiber.
7. In line 161 the authors should quantify the meaning of "considerable distance".

Reviewer #2 (Remarks to the Author):

Reviewer #3 (Remarks to the Author):

In this paper, the authors have devised and fabricated a metasurface consisting of silicon nanoposts with different diameters to implement beamsplitter/metalens with ~120degree field-of-view. By using their design in conjunction with a fiber array, the authors demonstrate an optical wireless communication scheme capable of collimating and deflecting/steering light to different angles. Moreover, they present a polarization sensitive metasurface that can steer light of different polarizations to different angles to simultaneously provide full-coverage for optical wireless communication and avoid cross-talk.

Generally, the paper is presented well and easy to follow. The measurements and theoretical framework are appropriate and thorough and the supplementary videos showcase the functionality of the device very clearly. However, some important references such as the one below are missing and not cited.

- He, Nan, et al. "Highly Compact all-solid-state beam steering module based on a metafiber." ACS Photonics 9.9 (2022): 3094-3101.

One of my main concerns is that the principle of operation (fiber array with metasurface composed of silicon nanoposts) and fabrication steps of the metasurface - which are the main contributions of this paper - are virtually identical to the above paper. Although the measurements and the application (free-space optical communication) are different and at a larger scale (i.e 12x12 fiber array and WDM), the rest of the system is based on conventional benchtop setups and doesn't introduce any particularly novel elements.

I think the polarization sensitive metasurface is a more interesting design, however, metasurfaces that deflect different polarizations of light to different angles (based on the same concept but in the visible range) have also previously been demonstrated (see below)

- Zhu, Wei, et al. "Polarization-multiplexed silicon metasurfaces for multi-channel visible light modulation." Advanced Functional Materials 32.22 (2022): 2200013.

Additionally, there are many typos and grammatical errors in the manuscript which need to be addressed.

Below are some comments and questions regarding the manuscript:

Main paper comments/questions:

1. In Fig.1c, can you please specify at what wavelength these simulations are done.
2. In Fig 1c. "No. of meta atoms" on the x-axis label can be confusing or misinterpreted. I'm guessing this refers to a meta atom/nanopost with a distinct structural parameter? (perhaps use index instead)
3. With respect to Line 194 and Fig 3: It's not clear, are all the wavelengths modulated with the same PRBS signal? As it stands, the wavelengths are spaced at around 800pm (~100GHz). With 25Gbps data-rate, the tail of adjacent channels will overlap which can deteriorate BER if the PRBS signals are different (i.e since the PRBS spectrum is sinc shaped, there will be significant signal power beyond 50GHz on either side of the carrier wavelength). If the PRBS signals are the same for adjacent channels, then your measurements will not capture this effect. In other words, the transmitted data must be uncorrelated (or you could use different fiber lengths for different channels to create timing difference). Can you please comment on this?
4. On line 353 you've stated 10dB improvement in SNIR when using polarization multiplexed metasurface. Is this 10dB improvement only due to employing a polarizer before the BERT? What would the SNIR be without the polarizer for the polarization sensitive metasurface?
5. In your measurements involving WDM, you have limited the wavelength range to 5nm. Is this because of chromatic aberration introduced by metasurface? For the designed metasurfaces, how does the measured deflection angle change vs wavelength over larger wavelength ranges (e.g 1540nm-1560nm as in your measurements related to Fig2f)?
6. Related to above, in Line 172 you state that 1560.61nm and 1539.77nm are the boundary wavelengths used in WDM communication characterization, but the actual wavelength ranges demonstrated in Fig. 3 and 4 are only 1546.92 to 1552.52nm. Can you please clarify?
7. Fig 5, the 3 panels at the bottom are not labeled and without captions.

There are numerous typos and grammatical errors in the paper. I've highlighted some of these below.

Main paper typos:

1. Fig 2d: y-label should read beam radius.
2. Caption of Fig 3b states that the wavelengths are spaced at 25GHz. This seems to be a typo.
3. Fig 6d: units of y-label should be dB for normalized optical power.
4. Line 118: independent of z? Shouldn't this read independent of r instead?
5. Line 191: Technically it is 25Gbits and not 25GHz

6. Line 256: typo "wide-filed" ... perhaps you meant wide field-of-view.
7. Line 353: should read "dB"

Supplementary comments/questions:

1. Supplementary Fig 1a: In the experimental setup to measure divergence of the beam from a SMF fiber, why is the metasurface included between fiber facet and beam profiler? Wouldn't the metasurface collimate the light?

Supplementary typos:

1. Line 52: "moved away"
2. Line 80: should read "for verification"
3. Line 151: "Comparison" should read "comparison"
4. Supplementary Fig1c: double check y-axis label, should read radius instead of diameter.

Reviewer #4 (Remarks to the Author):

Reviewer #5 (Remarks to the Author):

The manuscript by Wu et al. titled "Tbps wide-field parallel optical wireless communications based on a metasurface beam splitter" introduces an innovative optical wireless communication (OWC) scheme utilizing a compact beam splitter comprising a metasurface and fiber array. This scheme enables parallel support for users across a wide angle range, achieving high-speed communication with an ultra-high communication capacity exceeding 28 Tbps for the entire system. The demonstration is interesting and solid. I am happy to recommend the publication of this work with the following issues to be addressed.

1. The authors claim that this work reach 'highest reported communication rate to date', it would be helpful for the reader to understand if the author gives a brief introduction of the highest reported communication rate before and if there are any conditional limits in this work to achieve more higher communication rate.
2. In the experiment, the deflection efficiency maintains greater than 40% until the angle exceeds 60°. Could the author briefly analyze how to further improve efficiency? For example, is it beneficial to improve efficiency by using more types of discrete units? In addition, is the current discretization of the unit cell into 16 types limited by the existing processing technology?
3. In the introduction, the authors stated, "Although these solutions can achieve dynamic deflection of beams to a certain extent, it is difficult to achieve arbitrary deflection of multiple beams, and even more difficult to achieve parallel independent information transmission for each beam." However, it is not entirely clear why achieving arbitrary deflection (a typical functionality) with dynamic metasurfaces is challenging, and what difference between the designed metasurfaces from traditional ones.

4. It would be beneficial for the authors to provide a detailed comparison with [Adv. Mater. 34, 2106080 (2022)], which involves larger dimensions, among other differences.

5. It would be valuable for the authors to provide a forward-looking perspective on the future of optical communication. This could include discussing anticipated challenges, potential applications, such as in which sectors optical communication is likely to be utilized, and considerations regarding losses. Such insights would enrich the discussion and provide readers with a broader context for understanding the significance and potential impact of the research.

6. This study introduces promising advancements in parallel communications, with potential applications in 6G networks. Recent studies, such as "Homeostatic neuro-metasurfaces for dynamic wireless channel management" in Science Advances (2022), and "Curving THz wireless data links around obstacles" in Communications Engineering (2024), provide valuable insights into the integration of metasurfaces with wireless technologies, offering pertinent references for further investigation.

Dear Reviewers,

First of all, we would like to thank you for reviewing our manuscript entitled “Tbps wide-field parallel optical wireless communications based on a metasurface beam splitter”. The comments are valuable and helpful for improving the manuscript’s quality and guiding for further investigations. We have revised the original paper based on constructive comments and suggestions. Below are the detailed replies. The revisions are highlighted by using red text in the manuscript.

Reply to Reviewer#1:

First of all, we sincerely thank the Reviewer’s positive comments on our manuscript:

“The manuscript presents an optical wireless communication (OWC) link scheme that aims to simultaneously provide high-speed, multi-user information transfer over a wide area with no blind spots. The authors realize this scheme by combining a fiber array and a beam steering metasurface, experimentally demonstrating 200 Gbps data transmission, 60° deflection, and complete range covering through polarization multiplexing, and a prototype OWC for high-definition video transmission. The results presented are an improvement to the state-of-the-art beam steering device’s deflection angle and data rate. At the same time, the approach permits independent and parallel information transmission with no blind pots. The manuscript is well structured, and the data presented is of good quality. There is sufficient information to reproduce the results, and the work is of interest to a broad audience.”

We have prepared a point-by-point response to the Reviewer’s comments and revised our manuscript accordingly.

Reviewers’ comments:

Major remarks:

1. The introduction states that a limitation of single metasurfaces for OWC is fixed-angle beam steering, which is presently mitigated by control mechanisms such as liquid crystals or optomechanics that allow dynamic beam steering. Dynamic beam steering is engaging in OWC as it provides information that can be transferred to moving users. Here, Fig. 6a shows information transfer to a walking person and a robotic vacuum cleaner with the proposed scheme. However, this work describes a single metasurface setup for which communication lines with moving users should not be possible. The authors should describe the mechanism by which connections to moving users can be established or revise Fig. 6a to avoid misleading readers on the capabilities of the proposed link scheme.

Reply: We thank the Reviewer for pointing out the important issue. In our work, the illumination position of each beam is fixed and cannot track moving objects in real-time. If a receiver moves to another area, it is necessary to switch to the beam located at the new position. To prevent potential misleading of readers caused by the content in Fig. 6a, in the revised version, we have adjusted the content of the figure, by replacing the walking person and the robotic vacuum with a stationary printer and a desktop computer. The revised Fig. 6a is shown as below.

Revised Fig. 6a

2. In line 169, the authors state that at 60° deflection angles, the beam spots retain high quality. However, Fig. 2e shows considerable deformation of the beam profile, and Fig. 2f shows the 60° beam to carry less than half the energy of the 0° beam. Therefore, the statement appears to contradict the data. These statements repeat in the discussion, as lines 368-369 subjectively state that “the metasurface in this work effectively accomplishes large-angle beam deflection with good beam quality, which also maintains relatively high efficiency”

Reply: We thank the Reviewer for this helpful comment. In the previous version, we referred to large-angle beams as high-quality beams, which might be inaccurate. What we actually intended to express is that the beam quality is “relatively good.” In terms of beam shape, the 60° beam still maintains collimated and does not diverge significantly. As for the efficiency, it still retains close to 40%, which is relatively good for large-angle beam modulation by metasurfaces. To avoid misleading, in the revised manuscript, we have changed the descriptions to “relatively good quality” and “fair efficiency”, which are more suitable. In lines 176-178 in the revised manuscript, the sentence is modified as: “*Notably, even at a deflection angle as large as 60°, the beam spot still maintains relatively good quality.*”. In lines 400-401, the sentence is modified as: “*Significantly, the metasurface in this work effectively accomplishes large-angle beam deflection with relatively good beam quality, which also maintains fair efficiency.*”.

Minor remarks:

1. In Equation 1, the quantity A_0 should be described in the text as the other values are.

Reply: We thank the Reviewer’s kind suggestion. In the revised manuscript, in line 109, we have added a description of A_0 : “*where, A_0 is the amplitude of the electric field,*”.

2. There is inconsistent notation between equations 1-2, the text of the manuscript, and the supplementary material when referring to \tan^{-1} and \arctan . A consistent notation should be used.

Reply: We thank the Reviewer’s kind suggestion. We have standardized all the notation of inverse tangent function to the form of “ \tan^{-1} ”, throughout the manuscript and supplementary material. In

the revised manuscript, equation (1) is modified as: “ $E(r, z) = A_0 \frac{\omega_0}{\omega(z)} \cdot e^{-\frac{r^2}{\omega^2(z)}} \cdot e^{-i \left[k_0 \left(z + \frac{r^2}{2R(z)} \right) - \tan^{-1} \left(\frac{z}{z_0} \right) \right]}$,”

In lines 123-124, we revised the text as: “*Since the terms $k_0 d_0$ and $-\tan^{-1}(d_0/z_0)$ are all constant independent of r ,*”. In the revised supplementary material, equation (S1) is modified as

$$E(r, z) = A_0 \frac{\omega_0}{\omega(z)} \cdot e^{-\frac{r^2}{\omega^2(z)}} \cdot e^{-i \left[k_0 \left(z + \frac{r^2}{2R(z)} \right) - \tan^{-1} \left(\frac{z}{z_0} \right) \right]}$$

3. In Equation 2, the quantity k_0 should be described in the text.

Reply: We thank the Reviewer’s kind suggestion. In the revised manuscript, we have added a description of k_0 in line 123: “*where, $k_0 = 2\pi/\lambda_0$ is the wavenumber of 1550 nm wavelength beam.*”.

4. In Equation 3, the quantity d should be described in the text.

Reply: We thank the Reviewer’s kind suggestion. In equation (3), “ d ” denotes part of the derivative notation “ d/dr ”, rather than a variable. Our notation might cause misleading without description. In the revised manuscript, we have added a description of “ d/dr ” in line 129: “*where, d/dr denotes the symbol for the derivative with respect to r .*”.

5. The left axis of Fig. 1c is labeled “Efficiency”; however, the caption and line 138 refer to the panel as plotting “transmittance”. Thus, the panel’s axis label should be changed.

Reply: We thank the Reviewer’s kind suggestion. In the revised manuscript, we have modified the label as “Transmittance” in Fig. 1c, to keep consistent with the text. The revised figure is shown as below.

Revised Fig. 1c

6. In line 145 the authors should quantify the efficiency of light coupling into the fiber.

Reply: Thanks for the Reviewer’s kind suggestion which makes our manuscript more rigorous. We conducted an additional experiment to test the coupling efficiency. Figure R1(a) shows the measured optical power (3.69 dBm) directly emitted from the optical module, while Figure R1(b) shows the measured optical power (3.07 dBm) emitted from a short fiber connected to the optical module. It gives rise to an efficiency of ~87%. In the revised manuscript, we have added the specific efficiency value, in lines 151-152: “*Light modules with different wavelengths were chosen as the light source and efficiently coupled into a fiber (experimentally measured efficiency is approximately 87%).*”.

Fig. R1: Measurements of (a) the optical power directly emitted from the optical module, and (b) the optical power emitted from a short fiber connected to the optical module.

7. In line 161 the authors should quantify the meaning of “considerable distance”.

Reply: We thank the Reviewer’s helpful suggestion. Experimental analysis shows that, even after a propagation distance of 1 m, the beam radius remains smaller than 5 mm. We believe the distance of 1 m appropriately represents the “considerable distance”. In the revised manuscript, we have added the description of “considerable distance”, in lines 167-169: *“In comparison, the light beam modulated by the metasurface maintains its shape even at considerable distances (after a propagation of 1 m, the beam radius remains smaller than 5 mm).”*.

Reply to Reviewer#2:

Reviewers’ comments:

Reply: We thank the Reviewer for his/her co-review of our manuscript.

Reply to Reviewer#3:

First of all, we sincerely thank the Reviewer’s positive comments on our manuscript:

“In this paper, the authors have devised and fabricated a metasurface consisting of silicon nanoposts with different diameters to implement beamsplitter/metalens with ~120degree field-of-view. By using their design in conjunction with a fiber array, the authors demonstrate an optical wireless communication scheme capable of collimating and deflecting/steering light to different angles. Moreover, they present a polarization sensitive metasurface that can steer light of different polarizations to different angles to simultaneously provide full-coverage for optical wireless communication and avoid cross-talk. Generally, the paper is presented well and easy to follow. The measurements and theoretical framework are appropriate and thorough and the supplementary videos showcase the functionality of the device very clearly.”

We have prepared a point-by-point response to the Reviewer’s comments and revised our manuscript accordingly.

Reviewers’ comments:

1. However, some important references such as the one below are missing and not cited.

- He, Nan, et al. "Highly Compact all-solid-state beam steering module based on a metafiber." *ACS Photonics* 9.9 (2022): 3094-3101.

One of my main concerns is that the principle of operation (fiber array with metasurface composed of silicon nanoposts) and fabrication steps of the metasurface - which are the main contributions of this paper - are virtually identical to the above paper. Although the measurements and the application (free-space optical communication) are different and at a larger scale (i.e 12x12 fiber array and WDM), the rest of the system is based on conventional benchtop setups and doesn't introduce any particularly novel elements.

I think the polarization sensitive metasurface is a more interesting design, however, metasurfaces that deflect different polarizations of light to different angles (based on the same concept but in the visible range) have also previously been demonstrated (see below)

- Zhu, Wei, et al. "Polarization-multiplexed silicon metasurfaces for multi-channel visible light modulation." *Advanced Functional Materials* 32.22 (2022): 2200013.

Reply: We appreciate the two important references suggested by the Reviewer, and thank the Reviewer for his/her insightful criticism and advice. Our detailed responses to the issues raised by the Reviewer are as follows:

First, our primary novelty lies in the realization of the metasurface-based OWC system prototype. This prototype fully leverages the ultra-light, ultra-thin, flexible design and polarization multiplexing characteristics of metasurfaces. Consequently, it effectively addressed the challenges in traditional OWC system, such as difficulty in achieving large-angle and multi-beam deflection and ensuring full coverage of communication areas. Therefore, even though some of the techniques used in the mentioned references share similarities to our work, the application of these techniques in our work is aimed at addressing specific issues in OWC. In addition, we have approached more systematically with these techniques towards the particular application scenario.

Regarding the first reference [He, N. et al. Highly Compact All-Solid-State Beam Steering Module Based on a Metafiber. *ACS Photonics* 9, 3094–3101 (2022).], the authors proposed a beam steering module by assembling a metalens onto the end face of a fiber array. What they demonstrated is the application potential of the beam steering module in light detection and ranging (LiDAR), with a 2×2 fiber array. The principle of our beam steering is similar with theirs, that is using metasurface to deflect the light emitted from different optical fibers to different angles. However, in their paper, their target application is LiDAR, so they only need to put the fiber array at the focal plane of metasurface, to realize collimated deflected beams. This is insufficient for achieving full coverage OWC as our work emphasized. So, in our work, we firstly conducted a detailed analysis of the optical field distribution emitted by the fibers. Then the modulated beam shapes at different distances are analyzed and manipulated to achieve the full coverage OWC condition. Moreover, we have also carried out comprehensive researches on modulation of light with different wavelengths and polarizations by metasurface, in order to achieve higher communication speed or lower crosstalk. Therefore, our work leverages the multidimensional degrees of freedom in light modulation by metasurfaces.

Regarding the second reference [Zhu, W. et al. Polarization-Multiplexed Silicon Metasurfaces for Multi-Channel Visible Light Modulation. *Adv. Funct. Mater.* 32, 2200013 (2022).], the authors implemented polarization-multiplexed metasurfaces to achieve independent control of orthogonal linearly polarized light waves for multi-channel functionalities, including polarized beam splitting,

opposite charged and different ordered optical vortices. In fact, polarization multiplexing is a common method in multifunctional metasurface designs. Using polarization-multiplexed metasurface to deflect light with different polarizations to different angles is not surprise. However, most of literature works are only limited in function demonstration, and lacks addressing a specific application scenario. In our work, we employed the polarization-multiplexing to access a full indoor coverage for OWC with reduced communication crosstalk, fully reflecting the advantages of this design. Therefore, our study on polarization-multiplexed metasurface is target-oriented and quite efficient, being of significant innovation.

We have added the above two important references in the revised manuscript as references [41] and [43], and accordingly adjusted the numbering of other references. Additionally, we have included brief description of these two references as follows. In lines 97-100 of the revised manuscript: *“The assembling of fiber array and metasurface has been successfully implemented to realize collimated beam steering in light detection and ranging (LiDAR) ⁴¹. However, to achieve full area coverage in OWC, it is necessary to systematically analyze the light field emitted from the fiber and its modulation effect by the metasurface.”*. In lines 316-318 of the revised manuscript: *“To address this issue, we introduced the polarization-multiplexed metasurface, which would exhibit different phase responses when illuminated with opposite circularly polarized light ²⁶ or orthogonal linear polarized light ⁴³.”*.

The added references:

41. He, N. et al. Highly Compact All-Solid-State Beam Steering Module Based on a Metafiber. *ACS Photonics* 9, 3094–3101 (2022).

43. Zhu, W. et al. Polarization-Multiplexed Silicon Metasurfaces for Multi-Channel Visible Light Modulation. *Adv. Funct. Mater.* 32, 2200013 (2022).

Main paper comments/questions:

1. In Fig.1c, can you please specify at what wavelength these simulations are done.

Reply: Thank the Reviewer for pointing out the lack of clarity in our manuscript. The simulation results shown in Fig. 1c were conducted at wavelength of 1550 nm, which is the center wavelength of the beam steering device. In our dense wavelength division demultiplexing (DWDM) method, the wavelength range is very small (< 3.5 nm to the center wavelength), thus the difference in structural response caused by wavelength deviation can be neglected. In the revised manuscript we have added the simulation wavelength, in lines 142-145: *“Through finite difference time domain (FDTD) simulation, 16 kinds of nano-posts with different cross section radius were identified at the wavelength of 1550 nm, that cover 0-2 π phase range and all exhibit over 90% transmittance, as shown in Fig. 1c.”*. The figure caption of Fig. 1c is also modified, in lines 616-618: *“(c) Phase (brown stars) and transmittance (blue circles) of meta-atoms with 16 different structural parameters, simulated by FDTD solutions at the wavelength of 1550 nm.”*.

2. In Fig 1c. “No. of meta atoms” on the x-axis label can be confusing or misinterpreted. I’m guessing this refers to a meta atom/nanopost with a distinct structural parameter? (perhaps use index instead)

Reply: Thank the Reviewer for pointing out this issue. As the Reviewer mentioned, the x-axis label in Fig. 1c represents the meta atom with a distinct structural parameter. Using “Index of meta atoms”

is more appropriate. In the revised manuscript we have modified the x-axis label of Fig. 1c. The revised Fig. 1c is shown as below.

Revised Fig. 1c

3. With respect to Line 194 and Fig 3: It's not clear, are all the wavelengths modulated with the same PRBS signal? As it stands, the wavelengths are spaced at around 800pm (~100GHz). With 25Gbps data-rate, the tail of adjacent channels will overlap which can deteriorate BER if the PRBS signals are different (i.e since the PRBS spectrum is sinc shaped, there will be significant signal power beyond 50GHz on either side of the carrier wavelength). If the PRBS signals are the same for adjacent channels, then your measurements will not capture this effect. In other words, the transmitted data must be uncorrelated (or you could use different fiber lengths for different channels to create timing difference). Can you please comment on this?

Reply: We thank the Reviewer for pointing out this professional question. To answer in general, the PRBS signals modulated at different wavelengths are distinct, and in our work the impact of overlap between adjacent channels can be effectively reduced by the implementation of dense wavelength division multiplexing (DWDM) modules. The specific explanations are as follows.

Fig. R2 (a) The configuration of Quad transceiver on FPGA KCU116 Xilinx evaluation board, each transceiver is outlined by green dashed lines. (b) The block diagram of each transmitter (TX), the pattern generator block is circled in red.

Firstly, the FPGA KCU116 Xilinx evaluation board employed in experiment hosts four zSFP (z small form-factor pluggable) module connections and features four transceivers for driving the modules as shown Fig. R2(a). To drive the eight wavelengths optical modules, two FPGA boards were used in our work. The pattern generator block of each transceiver can be controlled independently to generate PRBS sequence, as shown in Fig. R2(b). As the resets of pattern generators are all asynchronous, and the two FPGA boards have independent clock, there would be timing differences among all the eight PRBS sequences. Thus, all the eight PRBS signals with different wavelengths are uncorrelated.

Secondly, as the Reviewer mentioned, the overlapping tails of different PRBS signals may contribute to the deterioration of BER. However, in our work, due to the implementation of DWDM modules (DWDM-SFP25G-10, Cisco), the interference caused by the overlapping of adjacent channels is negligible. Figure R3 shows the spectral profiles of PRBS signals carried by three DWDM modules with adjacent wavelengths of $\lambda_1=1549.32$ nm, $\lambda_2=1550.12$ nm, and $\lambda_3=1550.92$ nm, respectively. It can be observed that, each DWDM optical modules has a narrow spectra width. Then the three signals would be combined by the wavelength division multiplexer (MUX, FMU-D402160M3, FS), and next filtered by the demultiplexer (DMUX, FMU-D402160M3, FS). The two dashed lines in Fig. R3 represent the filter bandwidth of the DMUX, which has been experimentally tested to be 0.8 nm (test is shown in the newly added Supplementary Note S4 below). The portions of the adjacent channel signals within the dashed lines will cause interference to the middle signal. However, it is observed that the signal-to-interference ratio (SIR) exceed 40dB. Therefore, the impact of adjacent channel interference caused by the overlapping tails to BER is negligible.

Fig. R3: The spectral profiles of the optical signal output from the DWDM optical modules. The bonds of the wavelength division DUMX channel and the peak of the signal spectrum are indicated by brown dashed lines.

We believe the Reviewer’s question is important and meaningful. Therefore, we have added a discussion on this topic in the revised supplementary material as Supplementary Note S4, and updated the numbering of subsequent contents in the supplementary materials. In lines 239-240 of revised manuscript, we added the reference of this section: “*A detailed discussion on PRBS signals at different wavelength channels is provided in Supplementary Note S4.*”. In lines 206-243 of revised supplementary material: “

Note S4 PRBS signals at different wavelength channels

The FPGA KCU116 Xilinx evaluation board employed in experimental setup hosts four zSFP (z small form-factor pluggable) module connections. Thus, the board features four transceivers for driving the modules as shown in Fig. S6(a). To drive the eight wavelengths optical modules, two FPGA boards with independent clock were used in our work. Due to the asynchronism of different boards, the PRBS sequences generated by different boards are uncorrelated. Moreover, for each FPGA board, the pattern generator blocks of each transceiver can be controlled independently to generate PRBS sequence, as shown in Fig. S6(b). The resets of pattern generators are all asynchronous. Thus, there would be timing differences among the PRBS sequences generated by the same board. As a result, all of the eight PRBS sequences are uncorrelated.

Supplementary Figure S6 | Configuration of the FPGA board. (a) The configuration of Quad transceiver on FPGA KCU116 Xilinx evaluation board, each transceiver is outlined by green dashed lines. (b) The block diagram of each transmitter (TX), the pattern generator block is circled in red.

The overlapping tails of different PRBS signals may contribute to the deterioration of BER. However, due to the implementation of DWDM modules (dense wavelength division multiplexing, DWDM-SFP25G-10, Cisco) and the wavelength division multiplexer/demultiplexer (MUX/DMUX, FMU-D402160M3, FS), the interference from adjacent channels is limited. To illustrate the interference between adjacent channels, three PRBS-31 signals carried by wavelengths of $\lambda_1=1549.32$ nm, $\lambda_2=1550.12$ nm, and $\lambda_3=1550.92$ nm were combined by the wavelength division MUX and then demultiplexed by the wavelength division DMUX. The spectral profiles of the optical signal output from the optical module (Fig. S7(a)), wavelength division MUX (Fig. S7(b)) and the wavelength division DMUX (Fig. S7(b)) were recorded by the spectrometer (AQ6375, YOKOGAWA), as shown in Fig. S7. In Fig. S7(a), the peak of signal spectrum and the lower and upper bounds of the wavelength division channel are demarcated by brown dashed lines. The bandwidth of the DWDM channel is constrained within 0.8 nm as shown in Fig. S7(c). It can be seen from Fig. S7(a) that, the DWDM optical modules possess narrow spectral width. Thus, even though the tail of adjacent channel slight overlaps to the channel bandwidth of the signal, the signal-to-interference ratio (SIR) of adjacent channels still exceeds 40 dB. Therefore, the impact of the adjacent channel interference

caused by the overlapping tails to bit error rate (BER) is limited. In summary, for different PRBS signals, although there will be overlap in the tails of adjacent WDM channels, the impact of adjacent channel interference is minimal, allowing for the relatively satisfactory bit error rate (BER).

Supplementary Figure S7 | The spectral profiles of the optical signal output from (a) the optical module, (b) wavelength division MUX and (c) the wavelength division DMUX.

”.

4. On line 353 you’ve stated 10dB improvement in SNIR when using polarization multiplexed metasurface. Is this 10dB improvement only due to employing a polarizer before the BERT? What would the SNIR be without the polarizer for the polarization sensitive metasurface?

Reply: We thank the Reviewer’s question. Let us provide a more comprehensive discussion on this important issue. The 10dB improvement in SNIR not only attributes to employing a polarizer before the BERT, it is also associated with the beam spots distributions under different conditions. That is, to meet the condition of full range coverage, each beam modulated by non-polarization multiplexed metasurface has a larger divergence angle than the beam modulated by polarization multiplexed metasurface, as shown in Fig. R4(a) and (c). Such a larger divergence angle not only diminishes the received signal energy but also substantially elevates interference from adjacent channels, consequently contributing to the degradation of SNIR.

Fig. R4: Beam distributions and the corresponding experimental received signals spectrum of (a, b) non-polarization multiplexed metasurface method, (c, d) polarization multiplexed method without polarizer filtering, and (e, f) polarization multiplexed method with polarizer filtering. The working signal beam regions are marked with yellow circular curves. The red circles in (c, e) represent the LCP beam coverage regions, while the blue circles represent the RCP beam coverage regions.

To clearly illustrate the comparison of SNIR in the polarization multiplexed metasurface method with and without polarization filtering, we provide the schematic diagrams of the beam spots distributions (Fig. R4(c) and (e)) and the measured signal spectra (Fig. R4(d) and (f)), in these two cases. A working signal with carrier wavelength of $\lambda_s=1550.12$ nm was coupled into the center fiber, and four interference signals with carrier wavelengths of $\lambda_{I1}=1548.51$ nm, $\lambda_{I2}=1549.32$ nm, $\lambda_{I3}=1550.92$ nm, and $\lambda_{I4}=1551.72$ nm were coupled into the four adjacent fibers, respectively. A receiver is positioned behind the metasurface, to capture all the signals and transmit them to a spectrometer for analysis. The input working signal and interference signals were all set as liner polarized (LP). In this case, after passing through the metasurface, there would be both left-handed circular polarization (LCP) modulated beams and right-handed circular polarization (RCP) modulated beams, as shown in Fig. R4(c). Under this condition, the LCP and RCP beam spots surrounding the working signal beam spot will all interfere with the working signal. Although, the

center-to-center distances between the signal beam spots and the LCP beam spots (red circles in Fig. R4(c)) of λ_{I1} , λ_{I2} , λ_{I3} , and λ_{I4} are all the same, **while the RCP beam spots (blue circles in Fig. R4(c)) of λ_{I3} and λ_{I4} are much closer to the working signal beam spots than the RCP beam spots of λ_{I1} and λ_{I2}** . Thus, as the received spectrum without the polarization filtering shown in Fig. R4(d), the interference caused by λ_{I3} and λ_{I4} is greater than that caused by λ_{I1} and λ_{I2} .

For the Reviewer's second question, without polarization filtering, the SNIR for λ_{I3} and λ_{I4} is about 15dB, while for λ_{I1} and λ_{I2} is about 22dB. Figure R4(e) shows the beam distribution after polarization filtering, in which all the RCP beams are effectively filtered. Under this condition, only the four LCP beams surrounding the working signal beam region attribute to the interference. Figure R4(f) shows the corresponding received spectrum, exhibiting an SNIR of about 25dB, which is the same with the solid brown line in Fig. 6d. It is a 10dB SNIR performance improvement compared with that in non-polarization multiplexed metasurface method (as shown in Fig. R4(b)).

In the previous version, we did not clearly describe the polarization states of the beams emitted from fibers and modulated by the metasurface. In the revised manuscript we have modified the related contents to make them clear. In lines 370-376: *“For polarization multiplexed metasurface, the incident beams from fibers were all set as liner polarized (LP) light. Then the incident beams would be modulated to both left-handed circular polarization (LCP) beams and right-handed circular polarization (RCP) beams by the metasurface. The working signal beam was set as an LCP beam, thus the LCP and RCP beams surrounding the working signal beam region would all contribute to the interference signals. When the signals were received, an LCP polarizer could be employed to effectively filter part of the interference signals.”*.

To provide a more comprehensive comparison, we have also revised Fig. 6d by adding the received spectrum of polarization multiplexed metasurface method without filtering. The revised Fig. 6d is shown below. In lines 376-379, we revised the description of Fig. 6d: *“Figure 6d shows the received spectra of non-polarization multiplexed metasurface (solid brown line) method, polarization multiplexed metasurface method without polarization filtering (dashed blue line) and with polarization filtering (solid blue line), respectively.”*. In lines 682-687, we revised the figure caption of Fig. 6d: *“(d) Received spectra of non-polarization multiplexed metasurface method (solid brown line, abbreviated as NPM-MS), polarization multiplexed metasurface method without polarization filtering (dashed blue line, abbreviated as PM-MS) and with polarization filtering (solid blue line, abbreviated as PM-MS-F), which were all normalized by the working signal power. λ_s is 1550.12 nm, λ_{I1} - λ_{I4} are 1551.72nm, 1550.92nm, 1549.32nm and 1548.51nm, respectively.”*.

Revised Fig. 6d

Additionally, we believe that it is meaningful to add the detailed comparison of the SNIR performances of non-polarization multiplexed metasurface method and the polarization multiplexed metasurface method with and without polarization filtering. Therefore, we have added this topic in the revised supplementary material as Supplementary Note S8. In lines 388-391 of the revised manuscript, we added the reference of this section: “A detailed comparison of the SNIR performances of non-polarization multiplexed metasurface method and polarization multiplexed metasurface method with and without polarization filtering are shown in **Supplementary Note S8**.”. In lines 471-512 of the revised supplementary material: “

Note S8 Detailed SNIR performances comparison of non-polarization multiplexed metasurface method and polarization multiplexed metasurface method

A working signal with carrier wavelength of $\lambda_s=1550.12$ nm was coupled into the center fiber, and four interference signals with carrier wavelengths of $\lambda_{11}=1548.51$ nm, $\lambda_{12}=1549.32$ nm, $\lambda_{13}=1550.92$ nm, and $\lambda_{14}=1551.72$ nm were coupled into the four adjacent fibers, respectively. When using a non-polarization multiplexed metasurface to achieve full range coverage of the communication area, the distribution of the working signal beam and interference signal beams is shown in Fig. S16(a). Behind the metasurface, a receiver is positioned at the working signal beam region (marked by yellow circular curves) to capture all the signals and transmit them to a spectrometer for analysis. The distance between the metasurface and the fiber array under this condition is $d_1=1650$ μm . The received spectrum normalized to working signal is shown in Fig. S16(b), which is the same with the solid blue line in Fig. 6d, showing a signal to noise plus interference ratio (SNIR) of about 15dB.

When using a polarization multiplexed metasurface to achieve full range coverage, the input working signal and interference signals were all set as linear polarized (LP). In this case, after passing through the metasurface, there would be both left-handed circular polarization (LCP) modulated beams and right-handed circular polarization (RCP) modulated beams, as shown in Fig. S16(c). The distance between the metasurface and the fiber array is $d_1=1430$ μm . Under this condition, the LCP and RCP beams surrounding the working signal beam region will all interfere with the working signal. Figure S16(d) shows the received spectrum under this condition without the polarization filtering, from which it can be observed that the interference caused by λ_{13} and λ_{14} is greater than that caused by λ_{11} and λ_{12} . This is mainly because the RCP beams of λ_{13} and λ_{14} are closer to the working signal beam region. The SNIR for λ_{13} and λ_{14} is about 15dB, while for λ_{11} and λ_{12} is about 22dB.

Figure S16(e) shows the beam distribution after LCP polarizer filtering, in which all the RCP beams are effectively filtered. Under this condition, only the four LCP beams surrounding the working signal beam region attribute to the interference. This interference situation is somewhat similar to the case in the non-polarization multiplexed metasurface method, as shown in Fig. S16(a). However, in this case, the distance between the four interference signal beams regions and the working signal beam region is greater, resulting in a reduced impact of the interference signals on the working signals. Figure S16(f) shows the received spectrum with the polarizer filtering, exhibiting an SNIR of about 25dB, which is the same with the solid brown line in Fig. 6d. It is a 10dB SNIR performance improvement compared with that in non-polarization multiplexed metasurface method.

Supplementary Figure S16 | Beam distributions and the corresponding experimental received signals spectrum, of (a, b) non-polarization multiplexed metasurface method, (c, d) polarization multiplexed method without polarizer filtering, and (e, f) polarization multiplexed method with polarizer filtering. The working signal beam regions are marked with yellow circular curves. The red circles in (c, e) represent the LCP beam coverage regions, while the blue circles represent the RCP beam coverage regions. λ_s is 1550.12 nm, λ_{I1} - λ_{I4} are 1551.72nm, 1550.92nm, 1549.32nm and 1548.51nm, respectively.

5. In your measurements involving WDM, you have limited the wavelength range to 5nm. Is this because of chromatic aberration introduced by metasurface? For the designed metasurfaces, how does the measured deflection angle change vs wavelength over larger wavelength ranges (e.g 1540nm-1560nm as in your measurements related to Fig2f)?

Reply: We would like to thank the Reviewer for this valuable question. The wavelength range of WDM is indeed limited by the chromatic aberration of the metasurface. Light of different wavelengths will be modulated to different deflection angles by the metasurface, causing a shift on the receiving surface after traveling a certain distance L . Since the size of the receiving lens is fixed, the wavelength range of WDM will determine the communication distance L , as shown in Fig. R5(a).

Thus, the 5nm wavelength range of WDM is determined by a comprehensive consideration of communication distance and the number of wavelengths used in WDM.

The relationship between the beam deflection angle and the incident beam wavelength is

$$\sin\theta(r, \lambda) = -\frac{\lambda}{\lambda_0} \cdot \frac{r}{d_0 \left[1 + \left(\frac{\pi \omega_0^2}{\lambda d_0} \right)^2 \right]},$$

where λ_0 is the center wavelength of 1550 nm, d_0 is the

distance between the metasurface and the fiber facet, ω_0 is the waist radius of the Gaussian beam emitted from the fiber, and r is the beam incident position on metasurface. (The detailed derivation of this equation is provided in the newly added **Supplementary Note S3**). Figure R5(b) shows the deflection angle as a function of illuminating position r , at wavelengths of 1550 nm, 1539.77 nm, and 1560.61 nm, respectively (inset is the zoom-in picture of the dashed box part). It can be observed that, as the illuminating position r on the metasurface increases, the deflection angle differences among these wavelengths also gets larger. Due to the small wavelength deviation from the center wavelength of 1550 nm, the deflection angle changes are also very small.

Fig. R5: (a) Geometric relationship of deflection angle changes at different wavelengths. It should be noted that, for clarity of illustration, some length proportions in the figure are not to scale. (b) Deflection angle as a function of illuminating position r , at wavelengths of 1550 nm, 1539.77 nm, and 1560.61 nm, respectively. Inset is the zoom-in picture of the dashed box part

The deflection angle deviation results in a shift of the beam spot r_{shift} at receiving plane of distance L , which can be expressed as: $r_{shift} = \tan(\theta(r, \lambda) - \theta_0(r, \lambda_0)) \cdot L$. To ensure sufficient beam energy

can be coupled through the receiving lens into the fiber, the beam spots shift of different wavelengths must be smaller than the receiving lens radius r_{lens} , which is 9.2 mm (PAF2A-18C,

Thorlabs) in our work. The relationship can be derived as $L \leq \frac{r_{lens}}{\tan(\theta(r, \lambda) - \theta_0(r, \lambda_0))}$. By

substituting the maximum size r of our metasurface into the equation, the maximum allowable communication distance L under different wavelength ranges can be determined. For the wavelength of 1539.77 nm and 1560.61 nm the maximum communication distance L is 0.80 m and 0.76 m, respectively. For the wavelength of 1546.92 nm and 1552.52 nm, the maximum L is 3.23 m and 2.66 m, respectively. **Hence, for relatively short-distance communication, the wavelength range can be specified as 1539.77 nm to 1560.61 nm, which can enhance the multiplexed wavelengths number for higher communication capacity. While, by shrinking the wavelength range to 1546.92-1552.52 nm, we can ensure both high communication speed and relatively long communication distance.**

Additionally, when designing the communication system, we have also considered whether it would be possible to use achromatic metasurface to achieve WDM over a wide wavelength range. However, according to the sophisticated designs of achromatic metalens [1. Shrestha, S. et al. Broadband achromatic dielectric metalenses. *Light Sci. Appl.* 7, 85 (2018); 2. Xiao, X. et al. Design and parametric analysis of the broadband achromatic flat lens. *Infrared Laser. Eng.* 49, 20201032 (2020)], it is almost impossible for our metasurface with relatively large numerical aperture (NA~0.7) and size (mm-scale) to achieve achromatic properties.

We think the Reviewer’s question is important and meaningful, we have added the detailed discussion on how the deflection angle varies with wavelength and how to determine the wavelength range for the communication system, in the revised supplementary material as **Supplementary Note S3**. In lines 200-206 of the revised manuscript, we added the reference of this section: “*The metasurface is designed for a working wavelength of 1550 nm. Due to the chromatic aberration of metasurface, when the metasurface modulates light deviates from the working wavelength, the deflection angle would change. Therefore, to use WDM for each communication channel, a comprehensive analysis of multiple factors is required, including the multiplexed wavelengths, deflection angles, and communication distance, in order to determine the optimal parameters of the communication system (for detailed analysis, see Supplementary Note S3).*”. In lines 131-204 of the revised supplementary material: “

Note S3 Influence of metasurface chromatic aberration on deflection angle

The electric field of light emitted from a fiber port can be theoretically approximated by a Gaussian beam model as shown in Eq. (1) in the main text. The phase profile just before the metasurface placed in front of the fiber with a distance of d_0 would be simplified as a quadratic phase form shown as: $\varphi_1(r) = kr^2/2R(d_0) + C$. Where $k = 2\pi/\lambda$ is the wave number of the light with wavelength of λ , $R(d_0, \lambda) = d_0 \left[1 + (\pi\omega_0^2/\lambda d_0)^2 \right]$ is the function of d_0 and λ . The phase distribution of the metasurface designed for $\lambda_0 = 1550$ nm can be expressed as $\varphi_{lens}(r) = -k_0 r^2/2R(d_0(\lambda_0))$.

Supplementary Figure S4 | Influence of metasurface chromatic aberration on deflection angle. (a) Simulated transmittance and phase shift of meta atoms of three wavelengths. (b) Deflection angle as a function of illuminating position r ; at wavelengths of 1550 nm, 1539.77 nm, and 1560.61 nm, respectively. (c) Geometric relationship of deflection angle changes at different wavelengths. It

should be noted that, for clarity of illustration, some length proportions in the figure are not to scale.

Optical response of the 16 nano-posts with different diameters were identified at the wavelength of 1539.77nm (solid line), 1550.12nm (dashed line) and 1560.61nm (dotted line) as shown in Fig. S4(a). The 16 nano-posts exhibits equivalent responsiveness to light at 1539.77nm and 1560.61nm as observed at 1550nm, which infers the consistent phase delay of the metasurface across different wavelengths within the bandwidth of the WDM OWC system. Thus, when the divergent beam passes through the metasurface at position r_0 , the phase distribution just after the metasurface can be written as:

$$\begin{aligned}\varphi(r) &= \varphi_{lens}(r) + \varphi_1(r) = -\frac{k_0 \cdot r^2}{2R(d_0, \lambda_0)} + \frac{k \cdot (r - r_0)^2}{2R(d_0, \lambda)} \\ &= \frac{(k \cdot R(d_0, \lambda_0) - k_0 \cdot R(d_0, \lambda)) \cdot r^2}{2R(d_0, \lambda_0) \cdot R(d_0, \lambda)} - \frac{k \cdot r_0 \cdot r}{R(d_0, \lambda)} + \frac{k \cdot r_0^2}{2R(d_0, \lambda)} \\ &= \frac{(k \cdot R(d_0, \lambda_0) - k_0 \cdot R(d_0, \lambda)) \cdot r^2}{2R(d_0, \lambda_0) \cdot R(d_0, \lambda)} - \frac{k \cdot r_0}{R(d_0, \lambda)} \cdot r + C.\end{aligned}\quad (S11)$$

Eq. (S11) consisted of a first-order term that influences the deflection angle of the emitted light, a second-order term that influences the divergence angle of the emitted light, and a constant term.

Submitting $R(d_0, \lambda) = d_0 \left[1 + (\pi\omega_0^2 / \lambda d_0)^2 \right]$ into the second-order term, the second-order term of Eq.

(S11) can be written as:

$$\begin{aligned}& \frac{(k \cdot R(d_0, \lambda_0) - k_0 \cdot R(d_0, \lambda)) \cdot r^2}{2R(d_0, \lambda_0) \cdot R(d_0, \lambda)} \\ &= \frac{r^2}{2} \cdot \frac{k \cdot (\pi\omega_0^2 / \lambda_0 d_0)^2 - k_0 \cdot (\pi\omega_0^2 / \lambda d_0)^2 + k - k_0}{1 + (\pi\omega_0^2 / \lambda_0 d_0)^2 + (\pi\omega_0^2 / \lambda d_0)^2 + (\pi\omega_0^2 / \lambda d_0)^2 \cdot (\pi\omega_0^2 / \lambda_0 d_0)^2}\end{aligned}\quad (S12)$$

Since $\pi\omega_0^2 / \lambda d_0 \ll 1$, $k - k_0 = 2\pi \cdot (\lambda_0 - \lambda) / \lambda_0 \cdot \lambda \ll 1$, the second-order term of Eq. (11) approaches zero for all the wavelengths used in the WDM OWC systems. It can be inferred that, the metasurface exhibits uniform capability of transforming a divergent light beam into a precisely collimated one within the bandwidth of the WDM OWC systems. The gradient of the phase of metasurface can be expressed as the phase gradient of oblique projection with deflection angle determined by the generalized Snell's law, which would be expressed as:

$$\begin{aligned}\sin \theta(r_0) &= \frac{1}{k} \frac{\partial}{\partial r} (\varphi(r)) \Big|_{r=r_0} = \frac{1}{k} \cdot \left(-\frac{r_0 \cdot k}{R(d_0, \lambda)} + \frac{(k \cdot R(d_0, \lambda_0) - k_0 \cdot R(d_0, \lambda)) \cdot r_0}{R(d_0, \lambda_0) \cdot R(d_0, \lambda)} \right) \\ &= \frac{1}{k} \cdot \left(-\frac{k_0 \cdot r_0}{R(d_0, \lambda_0)} \right) = -\frac{\lambda}{\lambda_0} \cdot \frac{r_0}{d_0 \left[1 + (\pi\omega_0^2 / \lambda d_0)^2 \right]}\end{aligned}\quad (S13)$$

The deflection angle of three wavelengths at different relative position between the fiber center and metasurface are plotted in Fig. S4(b). Due to the small wavelength deviation from the center wavelength of 1550 nm, the deflection angle changes are also very small.

However, when the communication distance is relatively long, the deflection angle changes cannot be ignored. The geometric relationship of the beam deflection angle, communication distance, and the receiver lens size is illustrated in Fig. S4(c). For laser with the wavelength of λ , the deviation in the deflection angle results in a shift of the light spot at the receiving plane located at a distance of L as $r_{shift} = \tan(\theta(r, \lambda) - \theta_0(r, \lambda_0)) \cdot L$. To ensure that sufficient energy can be

coupled through the receiver collimator into the fiber, the centers of light spots at different wavelengths must fall within the effective area of the coupling lens. To be precise, the offset of the light spot must be less than the radius of the receiver lens r_{lens} , which is 9.2mm (PAF2A-18C, Thorlabs) in our manuscript. The offset reaches its maximum value at maximum $r_0 = \sin(60^\circ) \cdot R(d_0, \lambda_0)$. Thus:

$$L \leq \frac{r_{lens}}{\tan(\theta(r_0, \lambda) - \theta_0(r_0, \lambda_0))}. \quad (S14)$$

For the wavelength of 1539.77nm and 1560.61nm the maximum L that ensure sufficient coupled energy are repressively 0.80m and 0.76m. For the wavelength of 1546.92nm and 1552.52nm the maximum L are repressively 3.23m and 2.66m, which meets the requirements of most indoor optical wireless communication systems.

Hence, for short-range communication, the boundary wavelength is specified as 1539.77nm and 1560.61nm to augment the number of wavelengths multiplexed in the OWC system to elevate high communication speed. While, for long-distance communication, the boundary wavelength is set at 1546.92nm and 1552.52nm to ensure the coupling efficiency of the receiver collimator and the communication performances of the OWC system.

Figure S5 | The measured BER-ORP (bit-error-ratio versus optical received power) curves for ten spaced wavelengths from 1539.77 to 1560.61 nm that carried 25Gbits signals.

Furthermore, we have supplemented 1539.77 nm and 1560.61 nm as the carrier wavelengths to the WDM communication characterization. Figure S5 shows the measured bit-error-ratio (BER) curves versus the optical received power (ORP) for the ten wavelengths (specifically: 1539.77nm, 1546.92 nm, 1547.72 nm, 1548.51 nm, 1549.32 nm, 1550.12 nm, 1550.92 nm, 1551.72 nm, and 1552.52 nm, 1560.61nm) at a deflection angle of 30° and a receive distance of 50 cm. The curve indicate that the system could maintain good communication performances when these ten wavelengths are multiplexed. Thus, boundary wavelengths used in WDM communication. It is necessary to clarify that, due to the limitations of the experimental conditions, only ten wavelengths were multiplexed. However, in practical applications, the number of wavelengths for reuse can be increased to 27 (from C21 1560.61nm to C47 1539.77nm) to ensure high-speed communication over short distances.

”.

6. Related to above, in Line 172 you state that 1560.61nm and 1539.77nm are the boundary wavelengths used in WDM communication characterization, but the actual wavelength ranges demonstrated in Fig. 3 and 4 are only 1546.92 to 1552.52nm. Can you please clarify?

Reply: We thank the Reviewer for pointing out this issue that we have not clarified clearly in previous version. The boundary wavelengths of 1539.77 nm and 1560.61 nm were determined by the wavelength division multiplexer/demultiplexer (MUX/DMUX, FMU-D402160M3, FS) we used in experiment. The MUX/DMUX provides multiplexing of 40 channels (C21-C60) with wavelengths ranging from 1529.55 nm (C60) to 1560.61 nm (C21). Due to the working wavelength was determined as 1550 nm, the boundary wavelengths were determined to be 1560.61 nm (C21) and 1539.77 nm (C47) for beam steering efficiency test, which forms the widest symmetrical band centered at 1550 nm allowed by our experimental device.

As we have discussed in the above question, the boundary wavelengths used in WDM would affect the communication distance. Therefore, when conducting communication experiments, we should take an overall consideration on factors of communication distance and communication rate. At last, we only selected the eight wavelengths closest to 1550 nm (specifically: 1546.92 nm (C38), 1547.72 nm (C37), 1548.51 nm (C36), 1549.32 nm (C35), 1550.12 nm (C34), 1550.92 nm (C33), 1551.72 nm (C32), and 1552.52 nm (C31)). It ensures that each channel achieves a high-speed communication rate of 200 Gbps while maintaining a maximum communication distance of over 2.6 m, which is sufficient for many indoor communication scenarios.

To clarify the boundary wavelength used in WDM communication more clearly, in the revised manuscript, we have modified Fig. 2f by adding the deflection efficiency of wavelengths 1546.92 nm and 1552.52 nm, and revised the description of Fig. 2f in lines 178-188: *“Figure 2f shows the deflection efficiencies of five wavelengths (1560.61 nm, 1552.52 nm, 1550.12 nm, 1546.92 nm, and 1539.77 nm) with respect to different deflection angles, which is defined as the ratio of deflected beam power to input beam power. The 1550.12 nm is the center wavelength used in the wavelength division multiplexing (WDM) communication in **Characterization of metasurface-based OWC system** section. The 1552.52 nm and 1546.92 nm wavelengths are the two boundary wavelengths for relatively long-distance (>2.6 m) WDM communication, while the 1560.61 nm and 1539.77 nm wavelengths are the two boundary wavelengths for relatively short-distance (<1 m) WDM communication (see **Supplementary Note S3** for detailed analysis). The results show that for all the five wavelengths, the deflection efficiency maintain greater than 40% until the angle exceeds 60°, ensuring high-speed OWC through WDM in a wide-angle range.”*. The figure caption of Fig. 2f has also been revised, in lines 631-633: *“(f) Measured deflection efficiency of five wavelengths (1560.61 nm, 1552.52 nm, 1550.12 nm, 1546.92 nm, and 1539.77 nm) at different beam steering angles.”*. In lines 212-215, we added the explanation about why 1546.92 nm to 1552.52 nm is chosen as the wavelength range in WDM communication experiment: *“The chosen wavelengths are the eight closest ones to the center wavelength of 1550 nm in WDM communication, which ensure that each channel achieves a high-speed communication rate while maintaining a relatively long communication distance.”*.

Revised Fig. 2f

7. Fig 5, the 3 panels at the bottom are not labeled and without captions.

Reply: We thank the Reviewer for pointing out this issue. In the revised manuscript we have labeled the 3 panels as Fig. 5e, Fig. 5f, and Fig. 5g, respectively, and rewritten the captions of Fig. 5. The revised Fig. 5 and its caption are shown as below.

Fig. 5 | The design of full range coverage metasurface-based OWC. (a) Geometric relations of light rays from adjacent fibers, at critical full range coverage condition. The red lines represent the rays from fiber $i-1$, while the grey lines represent the rays from fiber i . The two angles are the beam

deflection angle of line 1 from fiber $i-1$, and the beam deflection angle of line 3 from fiber i , respectively. The critical full range coverage condition is to make these two deflection angles equal. The schematics of the same polarized beam coverage on the observation plane at $d_2=10$ mm, when Δr in equation (5) is set equals to **(b)** p and **(c)** $\sqrt{2}p$, the corresponding distances between the metasurface and the fiber array are 1430 μm and 1650 μm , respectively. **(d)** The schematic of LCP and RCP beam coverage on the same observation plane, when Δr is set equals to p . **(e-g)** The beam intensity distributions on observation plane corresponding to the conditions in (b-d), respectively. The black wireframes in **(f)** show the boundaries of beams emitted from different fibers. The parts encircled by red wireframes and blue wireframes in **(g)** are the beam coverage areas from LCP incident light and RCP incident light, respectively.

Additionally, corresponding modifications have also been made to the main text. In lines 306-313: *“Thus, if Δr in equation (5) is set as the interval of nearest neighbor fibers, that is $\Delta r = p$, the spots distribution of the fiber array would be shown like Fig. 5b. In this case, although the spots of adjacent fibers are connected, there would still be considerable areas on the observation plane, that are not covered by light spots, as shown in Fig. 5e. Only when Δr is set as the interval of second nearest neighbor fibers (as shown in Fig. 5c), that is $\Delta r = \sqrt{2}p$, the observation plane would be fully covered, and the spots of nearest neighbor fibers would overlap, as shown in Fig. 5f.”*. In lines 318-320: *“The phase for modulating fiber output light into vacant area in Fig. 5b is obtained by applying a translation of $p/2$ in both x and y directions to the quadratic phase of equation (4), which would be shown as,”*. In lines 322-323: *“By manipulating the polarization of fiber array output light (as shown in Fig. 5d), the metasurface can achieve complete indoor beam coverage, as shown in Fig. 5g.”*.

There are numerous typos and grammatical errors in the paper. I've highlighted some of these below.

Reply: We feel sorry for our carelessness. We appreciate the Reviewer's attention to the details of our manuscript and are grateful for his/her efforts in pointing out the specific errors. The comments are invaluable in improving the quality of our manuscript. We have addressed these errors one by one. We have also thoroughly checked the manuscript and corrected other minor typos and grammatical errors in red in the revised manuscript.

Main paper typos:

1. Fig 2d: y-label should read beam radius.

Reply: We thank the Reviewer for pointing out this error. In the revised manuscript, we have corrected the label as “beam radius”. The revised Fig. 2d is shown as below.

Revised Fig. 2d

2. Caption of Fig 3b states that the wavelengths are spaced at 25GHz. This seems to be a typo.

Reply: We thank the Reviewer for identifying the error in our manuscript. In fact, we intended to refer to eight wavelengths which spaced at 0.8 nm intervals and each carrying 25 Gbits signal. In the revised manuscript the caption of Fig. 3b has been modified. In lines 637-639: “**(b)** *The measured BER-ORP (bit-error-ratio versus optical received power) curves for eight spaced wavelengths from 1546.92 to 1552.52 nm each of which carries 25Gbits signals.*”.

3. Fig 6d: units of y-label should be dB for normalized optical power.

Reply: We thank the Reviewer for pointing out this error. In the revised manuscript, we have corrected the y-label of Fig. 6d as: “dB”. The revised Fig. 6d is shown as below.

Revised Fig. 6d

4. Line 118: independent of z? Shouldn't this read independent of r instead?

Reply: Thank the Reviewer for pointing out this error. In the revised manuscript, we have corrected this sentence. In lines 123-124 “*Since the terms $k_0 d_0$ and $-\tan^{-1}(d_0/z_0)$ are all constant independent of r,*”.

5. Line 191: Technically it is 25Gbits and not 25GHz.

Reply: Thank the Reviewer for pointing out this error. In the revised manuscript, we have corrected this sentence. In lines 208-212: “*Eight 25-Gbits dense wavelength division multiplexing optical*”.

modules (DWDM-SFP25G-10, Cisco) operating within wavelengths ranging from 1546.92 nm to 1552.52 nm (specifically: 1546.92 nm, 1547.72 nm, 1548.51 nm, 1549.32 nm, 1550.12 nm, 1550.92 nm, 1551.72 nm, and 1552.52 nm) were driven by a pseudo-random binary sequence generator from the evaluation board (FPGA KCU116, Xilinx).”.

6. Line 256: typo “wide-filed” ... perhaps you meant wide field-of-view.

Reply: Thank the Reviewer for pointing out this spelling mistake. In the revised manuscript, we have corrected this error. In lines 278-280: “*The comprehensive exhibition of the high-speed wide-field parallel OWC system and the detailed signal conversion processes during the information transmission, are shown in **Supplementary Movies S1-S3**.*”. Since “wide-field” and “wide field-of-view” essentially have the same meaning, we use “wide-field” here to maintain consistency with other parts of the manuscript.

7. Line 353: should read “dB”.

Reply: Thank the Reviewer for pointing out this error. In the revised manuscript, we have modified the text. In lines 382-383: “*which is more than 10dB improvement compared with that in non-polarization multiplexed metasurface method.*”.

Supplementary comments/questions:

1. Supplementary Fig 1a: In the experimental setup to measure divergence of the beam from a SMF fiber, why is the metasurface included between fiber facet and beam profiler? Wouldn't the metasurface collimate the light?

Reply: We thank the Reviewer for pointing out the error in Supplementary Fig. S1(a). It is the divergence testing of the beam emitted directly from the fiber facet. So it does not require the addition of a metasurface. We have removed the metasurface in Supplementary Fig. S1(a) in the revised version. The modified Supplementary Fig. S1(a) is shown as below.

Revised Supplementary Fig. S1(a)

Supplementary typos:

1. Line 52: “moved away”.

Reply: Thank the Reviewer for pointing out this typo. In lines 50-53 of the revised Supplementary material, we modified the text as: “*To evaluate the collimation of the beam, a beam profiler (NanoScan 2s Si/9/5) was mounted on a XYZ motorized stage and moved away from the fiber array facet with a step size of 0.5mm to measure the intensity profile of the laser, as illustrated in Fig. S1(a).*”.

2. Line 80: should read for “verification”

Reply: Thank the Reviewer for pointing out this spelling mistake. In lines 78-80 of revised Supplementary material, we modified the text as: “*In addition, this method was also utilized to*”.

estimate ω_0 of SMF-28 Ultra FC/PC Single Mode Patch Cables (Thorlabs P1-SMF28E-FC) for verification.”.

3. Line 151: “Comparison” should read comparison

Reply: Thank the Reviewer for pointing out this spelling mistake. We have corrected the errors in two places in the revised Supplementary material. In lines 301-303: *“Comparison of the approximate boundaries and the amplitude distributions calculated by angular spectrum method when $d=1650\mu\text{m}$, $d_2=10\text{mm}$, $\phi=45^\circ$ and (a) $\theta=0^\circ$, (b) 20° , (c) 40° and (d) 60° respectively.”.* In line 402: *“Supplementary Figure S14 | Comparison of received SIR.”.*

4. Supplementary Fig1c: double check y-axis label, should read radius instead of diameter.

Reply: Thank the Reviewer for pointing out the error in the y-axis label of Supplementary Fig. S1(c). In the revised Supplementary material, we have modified y-axis label as: “beam radius (μm)”. The modified Supplementary Fig. S1(c) is shown as below.

Revised Supplementary Fig. S1(c)

Reply to Reviewer#4:

Reviewers' comments:

Reply: We thank the Reviewer for his/her co-review of our manuscript.

Reply to Reviewer#5:

First of all, we sincerely thank the Reviewer's high evaluation of our manuscript:

“The manuscript by Wu et al. titled “Tbps wide-field parallel optical wireless communications based on a metasurface beam splitter” introduces an innovative optical wireless communication (OWC) scheme utilizing a compact beam splitter comprising a metasurface and fiber array. This scheme enables parallel support for users across a wide angle range, achieving high-speed communication with an ultra-high communication capacity exceeding 28 Tbps for the entire system. The

demonstration is interesting and solid. I am happy to recommend the publication of this work with the following issues to be addressed.”

We have prepared a point-by-point response to the Reviewer’s comments and revised our manuscript accordingly.

Reviewers’ comments:

1. The authors claim that this work reach ‘highest reported communication rate to date’, it would be helpful for the reader to understand if the author gives a brief introduction of the highest reported communication rate before and if there are any conditional limits in this work to achieve more higher communication rate.

Reply: We thank the Reviewer for the very helpful suggestion. First, we need to clarify that the highest communication rate refers to the comparisons made within the reported beam steering-based OWC research works. In Supplementary Table S1, we compared the performances of our work with previously reported beam steering-based OWC works across multiple metrics, including the communication rates. The table below is a concise version of Tables S1 for the Reviewer’s convenience. Through comparison, it can be found that the total communication capacity is related both to the communication rate per channel, and the numbers of parallel communication channels. In previously reported works, the highest transmission capacity was 80 Gbps per channel and 8.9 Tbps in total [J. Lightwave Technol. 36, 4486-4493 (2018)]. Our work not only achieves a higher communication rate per channel, but also supports more parallel channels, consequently, the overall communication capacity is higher than previously reported works.

In our work, the communication rate per channel has an upper limit, because the number of wavelengths that can be multiplexed per channel is constrained. The diffraction of different wavelengths on the metasurface has a certain deviation, thus wavelengths that deviate too much (>10 nm) from the central wavelength of 1550 nm would result in a significant decrease in reception efficiency. This limitation is explained in detail in the revised Supplementary Note S3. Theoretically, the number of communication channels can be increased without constraint, just by increasing the number of fibers. However, in practice, more communication channels require larger-size of fiber array and metasurfaces, which would be limited by the fabrication processes.

In the revised manuscript, in lines 451-453, we added the reference of Supplementary Table S1, and we revised the description of the “highest communication rate” by adding the qualifier of “beam steering-based OWC”, to make the statement more precise: “*which is the highest reported communication rate in beam steering-based OWC works to date. The detailed performance comparisons of our work and other relevant works are shown in Supplementary Table S1.*”. In lines 412-417, we added the discussion about the communication rate limit of our work: “*For the issue of communication rate limit in our work. The communication rate per channel has an upper limit because of the chromatic property of metasurface, which is explained in detail in Supplementary Note S3. Theoretically, the number of communication channels can be increased without constraint, just by increasing the number of fibers. However, in practice, more communication channels require larger-size of fiber array and metasurfaces, which would be limited by the processing technology.*”.

The concise version of Supplementary Table S1

Representative works	FOV	Channel number	Data rate 1 channel	Full coverage	Method
Our Work	120°	144	200 Gbps (28.8Tbps in total)	√	Metasurface
Adv. Mater. 34 , 2106080 (2022) From State Key Laboratory of Optical Communication Technologies and Networks, China	80°	14	10 Gbps	×	Metasurface
Nanophotonics. 12 , 3511–3518 (2023). From Peng Cheng Laboratory, China	20°	9	100 Gbps (900 Gbps in total)	×	Metasurface
Adv. Photonics Res. 4 , 2300127 (2023) From State Key Laboratory of Optical Communication Technologies and Networks, China	20°	9	10 Gbps	√	Metasurface & LCoS-SLM
Opt. Express 28 , 30851–30860 (2020) From Huazhong University of Science and Technology, China	6°	8	60 Gbps	×	LCoS-SLM
Laser Photonics Rev. 15 , 2000266 (2021) From Eindhoven University of Technology, Netherlands	35°	126	20 Gbps	√	Metasurface & AWGR
J. Light. Technol. 36 , 4486–4493 (2018) From Eindhoven University of Technology, Netherlands	37.2°	112	80 Gbps (8.9Tbps in total)	√	AWGR
IEEE Photonics Technol. Lett. 25 , 1428 (2013) From Vienna University of Technology, Austria	12°	1	3 Gbps	√	MEMS
IEEE Photonics Technol. Lett. 28 , 550 (2016) From RIT Technologies Ltd, Israel	4°	8	10 Gbps	×	MEMS
ACS Photonics 10 , 3052–3059 (2023) From Zhejiang University, China	80°	1	10 Gbps	√	Metalens & translation stage
Opt. Lett. 39 , 5427 (2014) From Eindhoven University of Technology, Netherlands	37.2°	1	10 Gbps	√	Tunable laser & Grating

Notes: LC-SLM: Liquid Crystal-Spatial Light Modulator, LCoS-SLM: Liquid Crystal on Silicon-Spatial Light Modulator, VCSEL: Vertical-Cavity Surface-Emitting Laser, AWGR: Arrayed Waveguide Grating Router, MEMS: Micro-Electrical Mechanical System.

2. In the experiment, the deflection efficiency maintains greater than 40% until the angle exceeds 60°. Could the author briefly analyze how to further improve efficiency? For example, is it beneficial to improve efficiency by using more types of discrete units? In addition, is the current discretization of the unit cell into 16 types limited by the existing processing technology?

Reply: We thank the Reviewer for pointing out this very important issue regarding the efficiency improvement of large-angle beam deflection. The reason for lower efficiency of large-angle beam deflection is that the periodically arranged, regular-shaped metasurface structures can only satisfy the phase modulation requirement for large-angle beam deflection, but cannot meet the amplitude modulation needed for high beam deflection efficiency.

Using more types of discrete units can help improve efficiency to some extent. Because the limited number of discrete structures is difficult to simultaneously meet the amplitude and phase modulation requirements for high-efficiency beam deflection of all angles. One effective way to improve the large-angle beam deflection efficiency involves structural optimization using

techniques such as topological optimization or deep learning. These approaches target the large-angle deflection efficiency as the optimization objective, and use the structure shapes as optimization parameters. Through the optimization process, continuous free-form structures capable of efficiently deflecting beams to large angles can be achieved [Sell, D. et al. Large-angle, multifunctional metagratings based on freeform multimode geometries. *Nano Lett.* 17, 3752–3757 (2017).]. However, this approach often requires optimizing the entire metasurface, which demands significant computational power when the metasurface size is large.

Another way to improve the large-angle beam deflection efficiency involves designing tilted metasurface structures tailored to each deflection angle, which can provide the spatial amplitude distribution needed for efficient large-angle beam deflection. [Wang, Y. et al. Metalens with tilted structures for high-efficiency focusing at large-angle incidences. *Chin. Opt. Lett.* 22, 053601 (2024).]. However, the titled metasurface structure is difficult to be fabricated and has poor structural stability, requiring further development of the processing technology.

Regarding the Reviewer's last question, as the unit structures we used in this work are cylindrical nano-posts with different diameters. The current 16 discrete parameter units have resulted in a structural size difference as small as only 6 nm between them, which is indeed the limit of our current processing technology. If the structure used includes other irregular structures, more types of discrete units can be fabricated. However, if the irregular structures have too fine details, it will also pose a challenge for the processing technology.

In the revised manuscript, we have added the detailed descriptions about how to further improve efficiency, and added the two aforementioned references. In lines 403-411: *“There are two main methods to improve the efficiency of large-angle beam deflection. One involves structural optimization using techniques such as topological optimization^{44,45} or deep learning⁴⁶⁻⁴⁸, targeting large-angle deflection efficiency as the optimization objective and using structure shapes or positions as parameters. This can result in free-form structures or non-periodic arrangements with high modulation efficiency for large-angle beams⁴⁹. The other type of method involves designing tilted metasurface structures tailored to each deflection angle to enhance efficiency⁵⁰. Both types of methods require comprehensive consideration of fabrication precision and structural stability, necessitating further systematic researches.”*

The added references:

49. Sell, D., Yang, J., Doshay, S., Yang, R. & Fan, J. A. Large-Angle, Multifunctional Metagratings Based on Freeform Multimode Geometries. *Nano Lett.* 17, 3752–3757 (2017).

50. Wang, Y. et al. Metalens with tilted structures for high-efficiency focusing at large-angle incidences. *Chin. Opt. Lett.* 22, 053601 (2024).

3. In the introduction, the authors stated, "Although these solutions can achieve dynamic deflection of beams to a certain extent, it is difficult to achieve arbitrary deflection of multiple beams, and even more difficult to achieve parallel independent information transmission for each beam." However, it is not entirely clear why achieving arbitrary deflection (a typical functionality) with dynamic metasurfaces is challenging, and what difference between the designed metasurfaces from traditional ones.

Reply: We thank the Reviewer's question and acknowledge that we may not explained this part clearly in previous version. What we intended to convey is that dynamic control of multiple beams is difficult, and the realization of parallel communication channels is more difficult. In the

metasurface based dynamic beam deflection works as we referred, most involve dynamically modulating a single input beam to a single output beam. For instance, as shown in Fig. R6(a), dynamic modulation of the output beam is achieved by rotating multilayer metasurfaces, and in Fig. R6(b), dynamic modulation is achieved by translating the single input fiber. However, it is difficult for these methods to achieve dynamic control of multiple output beams. The scheme based on the combination of metasurfaces and spatial light modulators (SLM) can achieve dynamic control of multiple output beams, as shown in Fig. R6(c). However, in this scheme, there is only one incident beam, so the multiple output beams cannot be used for independent information transmission and can only achieve a **broadcasting** information transmission. Compared with the aforementioned schemes, our scheme is a design with multiple input beams and multiple output beams, where the modulation of each beam is independent. This allows for parallel and independent information transmission, greatly increasing the communication capacity.

In the revised manuscript, we have adjusted this sentence to avoid ambiguity. In lines 70-73: *“These works modulate a single incident beam into one or multiple dynamically deflected beams. However, since these output beams all originate from one single incident beam, they are not independent of each other, thus cannot be used to achieve parallel communication functions.”*

Fig. R6: Schematics of three dynamic beam modulation schemes based on metasurfaces. (a) Ref. 39 [Sci. Adv. 9, eadf8478 (2023).]; (b) Ref. 40 [ACS Photonics 10, 3052–3059 (2023).]; (c) Ref. 38 [Adv. Photonics Res. 4, 2300127 (2023).].

4. It would be beneficial for the authors to provide a detailed comparison with [Adv. Mater. 34, 2106080 (2022)], which involves larger dimensions, among other differences.

Reply: We thank the Reviewer for the helpful suggestion. The reference mentioned is an excellent work on metasurface beam steering for OWC. Although this work and ours both achieve multiple output beam deflection, our work differs from this work in many aspects, including design principle, OWC scenarios, and final implementation performances etc. The specific differences are as follows:

- (1) Our beam steering system involves multi-beam input and multi-beam output, as shown in Fig. R7(a), whereas theirs involves single-beam input and multi-beam output, as shown in Fig. R7(c). The output beams in our system are independent of each other, but theirs are not.
- (2) In their system, the beam deflection modulation phase is obtained by a holographic method, with the modulation phases of all beams designed as a whole through the Gerchberg-Saxton (GS) algorithm. In our system, the beam deflection phases are obtained through analysis based on the generalized Snell's law, with each beam's deflection modulation phase analyzed and designed individually.
- (3) The metasurface they used is a reflective type, while ours is transmission type. Comparatively, transmission optical elements are more convenient for usage and assembly.

(4) Due to the independence of our multiple deflected beams, we can easily achieve multi-channel parallel communications. In their work, all the deflected beams are generated from one single incident beam, so their system is mainly used for **broadcasting communication**.

(5) In their work, wavelength division multiplexing (WDM) technology is also used to enhance data transmission capacity. However, in their system, light signals of different wavelengths are transmitted to different deflected beams, i.e. different receivers, as shown in Fig. R7(d). As a result, in their work, the communication rate for each channel can only reach 10 Gbps, with a total transmission capacity of up to 100 Gbps. In our work, WDM technology is used to increase the transmission capacity of each channel. Each of our channels can receive light signals with all the 8 multiplexed wavelengths, as shown in Fig. R7(b), allowing our communication rate for each channel to reach 200 Gbps. With 144 channels, the total transmission capacity can reach 28 Tbps.

(6) In their work, the deflected beams are all collimated narrow beams, which limit the coverage area for communication. In contrast, our work uses slightly divergent beams to cover the entire indoor area, ensuring communication without blind spots.

In the revised manuscript, in lines 190-195, we have added a comparison of our work and the mentioned reference: *“In previously reported studies, reflective metasurfaces designed on the principle of holography have been utilized to deflect multiple beams from one incident light for broadcasting in optical wireless communication (OWC) systems^{6,38}. However, in these works, all the deflected output beams carry identical information, unable to achieve parallel communication for multiple users. In our research, we adopt a different approach: loading different optical signals onto distinct fibers within an array.”*

Fig. R7: Schematics beam deflection and experimental setup in our work (a, b) and in the comparison work (c, d) [Adv. Mater. 34, 2106080 (2022)].

5. It would be valuable for the authors to provide a forward-looking perspective on the future of optical communication. This could include discussing anticipated challenges, potential applications, such as in which sectors optical communication is likely to be utilized, and considerations regarding losses. Such insights would enrich the discussion and provide readers with a broader context for understanding the significance and potential impact of the research.

Reply: We thank the Reviewer for the very constructive suggestion. In the revised manuscript, we have added a perspective on the future of optical wireless communications (OWC), including potential applications and challenges, etc. In lines 422-438: *“OWC features immune to electromagnetic interference, high level of privacy, easy installation, access to a huge license-free spectrum, and large communication capacity. These characteristics are advantageous in densely populated environments such as in open office room, conference halls, exhibition halls, airplane cabins, train cabins, etc⁵¹. By integrating with indoor high-precision optical sensing, the OWC technology will give rise to a series of intelligent indoor applications, such as health monitoring, fall detection, person identification, etc⁵². To achieve these functionalities, continuous developments are also needed in various technologies, such as light sources, beam modulation technology, target positioning technology, and optical signal receivers. Additionally, it requires comprehensive optimization methods for the layout of the entire communication system, energy distribution, and system losses. For example, intelligent real-time management of wireless channel according to communication demands⁵³, or solutions to the line-of-sight blockage problem between a base station and a user⁵⁴. Metasurfaces, with their ultra-light and ultra-thin advantages and flexible design capabilities, will have broad application prospects in the realization of highly integrated and high-performance optical communication devices, such as wide-angle optical signal receivers, high-precision positioning devices, and intelligent reflecting surfaces (RIS) for channel optimization⁵⁵.”*

The added references:

51. Koonen, T. Indoor Optical Wireless Systems: Technology, Trends, and Applications. *J. Light. Technol.* 36, 1459–1467 (2018).
52. K. Zhang, Z. Zhang, & B. Zhu. Beacon LED Coordinates Estimator With Selected AOA Estimators for Visible Light Positioning Systems. *IEEE Trans. Wirel. Commun.* 23, 1713–1727 (2024).
53. Fan, Z. et al. Homeostatic neuro-metasurfaces for dynamic wireless channel management. *Sci. Adv.* 8, eabn7905.
54. Guerboukha, H., Zhao, B., Fang, Z., Knightly, E. & Mittleman, D. M. Curving THz wireless data links around obstacles. *Commun. Eng.* 3, 58 (2024).
55. Aboagye, S., Ndjiongue, A. R., Ngatched, T. M. N., Dobre, O. A. & Poor, H. V. RIS-Assisted Visible Light Communication Systems: A Tutorial. *IEEE Commun. Surv. Tutor.* 25, 251–288 (2023).

6. This study introduces promising advancements in parallel communications, with potential applications in 6G networks. Recent studies, such as "Homeostatic neuro-metasurfaces for dynamic wireless channel management" in *Science Advances* (2022), and "Curving THz wireless data links around obstacles" in *Communications Engineering* (2024), provide valuable insights into the integration of metasurfaces with wireless technologies, offering pertinent references for further investigation.

Reply: We greatly appreciate the Reviewer for bringing these two works. These studies respectively address the issues of intelligent management of wireless channels and the blockage of communication channels by obstacles in wireless communications. Both issues are very important and practical in the real-world application of wireless communication. In the revised version, we mentioned these two issues in the discussion of the future of OWC and added these two papers as

references 53 and 54, respectively. We believe that these works provide valuable references for the development of our future work.

We hope that the above responses would be sufficient for Reviewers' comments and help to make our manuscript qualified for publication in **Nature Communications**.

Sincerely yours,
Ji Chen, Tao Li

National Mobile Communications Research Laboratory, School of Information Science and Engineering, Frontiers Science Center for Mobile Information Communication and Security, Southeast University, Nanjing 211189, China.

National Laboratory of Solid State Microstructures, College of Engineering and Applied Science, School of Physics, Nanjing University, Nanjing, 210023, China.

E-mail: jichen@seu.edu.cn;
taoli@nju.edu.cn;

REVIEWERS' COMMENTS

Reviewer #1 (Remarks to the Author):

The authors have addressed all of my comments with revisions to the manuscript text and figures. As a suggestion, the authors may consider citing the following works to complement their introductory discussion on state-of-the-art metasurface applications:

- <https://doi.org/10.1021/acsnano.3c09798>
- <https://doi.org/10.48550/arXiv.2312.10639> (Will appear in Nature Communications)
- <https://doi.org/10.1063/5.0204694>

Reviewer #2 (Remarks to the Author):

Reviewer #3 (Remarks to the Author):

I think the authors have done well to revise and modify the manuscript and have sufficiently answered my questions and concerns. The revision includes additional information that clarifies and expands on previously lacking details and has thus improved the manuscript's quality.

A few minor issues and typos to be addressed:

Line 184: two boundary wavelengths for relatively long-distance (>2.6m) ... Since 2.6m is the max distance, this should rather read $\sim 2.6\text{m}$ or $< 2.6\text{m}$

Line 371: Typo: should read "linear" instead of "liner"

Supplementary

Line 154 and Line 155: "influent" should read "influences" or impacts

Line 159: Some symbols don't appear correctly

Line 181 and 182: should read "respectively" instead of "repressively"

Line 200: "Thus, boundary wavelengths used in WDM communication" is not a complete sentence. Please revise

Line 485: "Linear"

Reviewer #4 (Remarks to the Author):

Reviewer #5 (Remarks to the Author):

The author has answered all the questions and I agree to its publication

Dear Reviewers,

Firstly, we would like to thank you for reviewing the revised version of our manuscript entitled “Tbps wide-field parallel optical wireless communications based on a metasurface beam splitter”. And we appreciate your agreement on its publication. We have revised our manuscript based on the constructive comments and suggestions from this round of peer review. Below are the detailed replies.

Reply to Reviewer#1:

Reviewers' comments:

The authors have addressed all of my comments with revisions to the manuscript text and figures. As a suggestion, the authors may consider citing the following works to complement their introductory discussion on state-of-the-art metasurface applications:

- <https://doi.org/10.1021/acsnano.3c09798>

- <https://doi.org/10.48550/arXiv.2312.10639> (Will appear in Nature Communications)

- <https://doi.org/10.1063/5.0204694>

Reply: We appreciate the Reviewer's feedback and are pleased to hear that the Reviewer find our revisions satisfactory. Regarding the Reviewer's suggestion to cite additional references, we have carefully reviewed these works and found that the first and third suggested works are relevant to our manuscript. Therefore, we have replaced the original references [23] and [24] with these two works. The updated two references are as follows.

[23] Barkey, M. *et al.* Pixelated High-Q metasurfaces for in situ biospectroscopy and artificial intelligence-enabled classification of lipid membrane photoswitching dynamics. *ACS Nano* **18**, 11644-11654 (2024).

[24] Schulz, S. A. *et al.* Roadmap on photonic metasurfaces. *Appl. Phys. Lett.* **124**, 260701 (2024).

Reply to Reviewer#2:

Reviewers' comments:

Reply: We thank the Reviewer for his/her co-review of our manuscript.

Reply to Reviewer#3:

First of all, we appreciate the Reviewer's positive feedback on our revised manuscript:

"I think the authors have done well to revise and modify the manuscript and have sufficiently answered my questions and concerns. The revision includes additional information that clarifies and expands on previously lacking details and has thus improved the manuscript's quality."

We have prepared a point-by-point response to the issues raised by the Reviewer in this round of peer review.

Reviewers' comments:

1. Line 184: two boundary wavelengths for relatively long-distance (>2.6m) ... Since 2.6m is the max distance, this should rather read ~2.6m or <2.6m

Reply: Thank the Reviewer for pointing out this issue, in the latest revised version, we have modified the expression to (~2.6 m) in Line 176.

2. Line 371: Typo: should read "linear" instead of "liner".

Reply: Thank the Reviewer for pointing out this typo, in the latest revised version, we have corrected this spelling mistake.

Supplementary

1. Line 154 and Line 155: "influents" should read "influences" or impacts.

Reply: Thank the Reviewer for pointing out this issue, in the latest revised Supplementary material, we have modified the words to "influences" in Line 154 and Line 155.

2. Line 159: Some symbols don't appear correctly.

Reply: Thanks for the Reviewer's kind reminder, in the latest revised Supplementary material, we have addressed this formatting issue, the equations are shown as: " $\pi\omega_0^2/\lambda d_0 \ll 1$,

$$k - k_0 = 2\pi \cdot (\lambda_0 - \lambda) / \lambda_0 \cdot \lambda \ll 1$$

3. Line 181 and 182: should read "respectively" instead of "repressively".

Reply: Thank the Reviewer for pointing out this error, in the latest revised Supplementary material, we have modified these spelling mistakes.

4. Line 200: "Thus, boundary wavelengths used in WDM communication" is not a complete sentence. Please revise.

Reply: Thank the Reviewer for pointing out this issue, in the latest revised Supplementary material, the sentence has been modified as: "Thus, boundary wavelengths used in WDM communication can be set as 1539.77 nm and 1560.61 nm."

5. Line 485: "Linear".

Reply: Thank the Reviewer for pointing out this error, we have modified this spelling mistake in the latest revised Supplementary material.

Reply to Reviewer#4:

Reviewers' comments:

Reply: We thank the Reviewer for his/her co-review of our manuscript.

Reply to Reviewer#5:

Reviewers' comments:

The author has answered all the questions and I agree to its publication.

Reply: Thanks for the Reviewer's satisfaction with our revisions and the agreement to the publication of our manuscript.